# CaRe-BN: Precise Moving Statistics for Stabilizing Spiking Neural Networks in Reinforcement Learning

**Zijie Xu, Xinyu Shi, Yiting Dong, Zihan Huang, Zhaofei Yu**[*]
Peking University
Beijing, 100871, China

## Abstract

Spiking Neural Networks (SNNs) offer low-latency and energy-efficient decision-making on neuromorphic hardware by mimicking the event-driven dynamics of biological neurons. However, the discrete and non-differentiable nature of spikes leads to unstable gradient propagation in directly trained SNNs, making Batch Normalization (BN) an important component for stabilizing training. In online Reinforcement Learning (RL), imprecise BN statistics hinder exploitation, resulting in slower convergence and suboptimal policies. While Artificial Neural Networks (ANNs) can often omit BN, SNNs critically depend on it, limiting the adoption of SNNs for energy-efficient control on resource-constrained devices. To overcome this, we propose Confidence-adaptive and Re-calibration Batch Normalization (CaRe-BN), which introduces (*i*) a confidence-guided adaptive update strategy for BN statistics and (*ii*) a re-calibration mechanism to align distributions. By providing more accurate normalization, CaRe-BN stabilizes SNN optimization without disrupting the RL training process. Importantly, CaRe-BN does not alter inference, thus preserving the energy efficiency of SNNs in deployment. Extensive experiments on both discrete and continuous control benchmarks demonstrate that CaRe-BN improves SNN performance by up to $22.6\%$ across different spiking neuron models and RL algorithms. Remarkably, SNNs equipped with CaRe-BN even surpass their ANN counterparts by $5.9\%$. These results highlight a new direction for BN techniques tailored to RL, paving the way for neuromorphic agents that are both efficient and high-performing. Code is available at `https://github.com/xuzijie32/CaRe-BN`.

## 1 Introduction

Spiking Neural Networks (SNNs) have emerged as a promising class of neural models that more closely mimic the event-driven computation of biological brains (Maass, 1997; Gerstner et al., 2014). This event-driven property makes SNNs particularly well suited for deployment on neuromorphic hardware platforms (Davies et al., 2018; DeBole et al., 2019), enabling low-latency and energy-efficient inference.

In parallel, Reinforcement Learning (RL) has achieved remarkable success across a wide range of domains (Mnih et al., 2015; Lillicrap et al., 2015; Haarnoja et al., 2018b). Among these, complex control tasks have received significant attention due to their alignment with real-world scenarios and their strong connection to embodied AI and robotic applications (Kober et al., 2013; Gu et al., 2017; Brunke et al., 2022). Combining the strengths of SNNs with RL (SNN-RL) offers the potential to train agents that not only learn complex behaviors but also execute them with extremely low energy consumption (Yamazaki et al., 2022). This makes SNN-RL particularly appealing for robotics and autonomous systems deployed on resource-constrained edge devices.

However, training SNNs is challenging. Due to the discrete spike dynamics and the reliance on surrogate gradients to approximate the backward pass, directly trained SNNs often suffer from unstable gradient propagation, including vanishing or exploding gradients (Zheng et al., 2021). Batch

---

[*]Corresponding author: yuzf12@pku.edu.cn

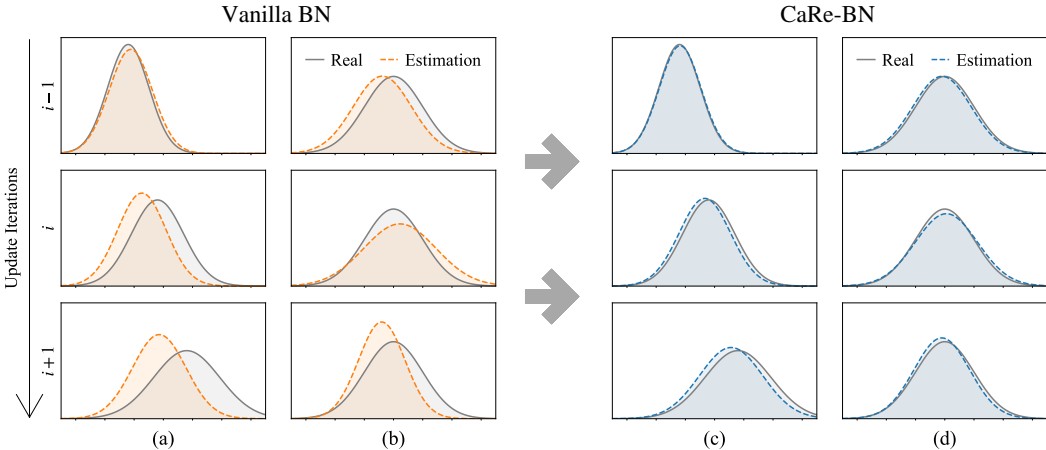

Figure 1: Real and estimated input activation distributions in BN layers. Between each gradient update iterations, distributions change rapidly in (a) and (c), while remaining stable in (b) and (d).

Normalization (BN) (Ioffe & Szegedy, 2015) plays a crucial role in stabilizing SNN training by regulating activation statistics and improving gradient flow, mitigates such instability and contributes to state-of-the-art performance (Duan et al., 2022; Jiang et al., 2024).

While effective in supervised learning, BN suffers a severe breakdown in online RL because moving statistics cannot be estimated precisely under nonstationary dynamics. As shown in Figure 1, traditional BN struggles to track the true statistics: When distributions shift rapidly (Figure 1(a)), estimates lag behind; when distributions are relatively static (Figure 1(b)), estimates contain noise. These inaccuracies lead agents to select suboptimal actions and generate poor trajectories, which are then reused for training—further compounding the problem and hindering policy improvement.

This issue is especially critical for SNNs. Traditional online RL algorithms usually remove BN layers in their networks (Sutton & Barto, 2018; Fujimoto et al., 2018; Haarnoja et al., 2017; Schulman et al., 2017). Unlike ANNs that can train stably without BN, SNNs rely heavily on normalization to stabilize membrane potentials and surrogate-gradient backpropagation. Removing BN from SNN-based RL leads to severe gradient instability and substantial performance degradation.

In this work, we address this issue by proposing **Confidence-adaptive and Re-calibration Batch Normalization** (CaRe-BN), a BN strategy tailored for SNN-based RL. CaRe-BN introduces two complementary components: (*i*) **Confidence-adaptive update** (Ca-BN), a confidence-weighted moving estimator of BN statistics that ensures unbiasedness and optimal variance reduction; and (*ii*) **Re-calibration** (Re-BN), a periodic correction scheme that leverages replay buffer resampling to refine inference statistics. Together, these mechanisms enable precise, low-variance estimation of BN statistics under the nonstationary dynamics of SNN-RL (Figure 1). With more accurate moving statistics, CaRe-BN stabilizes SNN optimization **without disrupting the online RL process**.

We evaluate CaRe-BN on a variety of control tasks, including the Atari benchmark (Bellemare et al., 2013; Machado et al., 2018) for discrete action spaces and the MuJoCo suite (Todorov et al., 2012; Todorov, 2014b) for continuous control. The results show that CaRe-BN not only resolves the issue of imprecise BN statistics but also accelerates training and achieves state-of-the-art performance. Remarkably, SNN-based agents equipped with CaRe-BN even **outperform their ANN counterparts by 5.9%**, without requiring complex neuron dynamics or specialized RL frameworks.

## 2    RELATED WORKS

### 2.1    BATCH NORMALIZATION IN SPIKING NEURAL NETWORKS

Batch Normalization (BN) was originally proposed for ANNs to mitigate internal covariate shift during training (Ioffe & Szegedy, 2015), thereby accelerating convergence and improving perfor-

mance (Santurkar et al., 2018). To address unstable training in SNNs, several extensions of BN have been developed (Zheng et al., 2021; Duan et al., 2022; Kim & Panda, 2021; Jiang et al., 2024). While these methods are effective in supervised tasks, they are designed under the assumption of static distributions. This assumption is violated in online RL, where distributions shift continually as the agent interacts with the environment, making these BN variants ill-suited for SNN-RL.

## 2.2 SPIKING NEURAL NETWORKS IN REINFORCEMENT LEARNING

Early work in SNN-RL primarily relied on synaptic plasticity rules, particularly reward-modulated Spike-Timing-Dependent Plasticity (R-STDP) and its variants (Florian, 2007; Frémaux & Gerstner, 2016; Gerstner et al., 2018; Frémaux et al., 2013; Yang et al., 2024). Another research direction focused on ANN-to-SNN conversion: while Patel et al. (2019); Tan et al. (2021); Kumar et al. (2025) converted Deep Q-Networks (DQNs) (Mnih, 2013; Mnih et al., 2015) into SNNs for discrete control, recent work by Xu et al. (2026) has greatly reduced conversion errors in continuous control. To enable direct gradient-based training, Liu et al. (2022); Chen et al. (2022); Qin et al. (2022); Sun et al. (2022) applied Spatio-Temporal Backpropagation (STBP) (Wu et al., 2018) to train DQNs, while Bellec et al. (2020) introduced e-prop with eligibility traces to train policy networks using policy gradient methods (Sutton et al., 1999).

For continuous control tasks, hybrid frameworks have been extensively explored (Tang et al., 2020; 2021; Zhang et al., 2022; Chen et al., 2024a; Zhang et al., 2024; Ding et al., 2022; Chen et al., 2024b). These approaches typically employ a Spiking Actor Network (SAN) co-trained with a deep ANN critic in the Actor–Critic framework (Konda & Tsitsiklis, 1999). Empowered by the recent proxy target framework (Xu et al., 2025), simple SNNs are now capable of outperforming their ANN counterparts. However, none of these methods address the challenge of normalization in SNN-based RL. The absence of proper normalization often leads to unstable updates and slower convergence.

# 3 PRELIMINARIES

## 3.1 SPIKING NEURAL NETWORKS

Spiking Neural Networks (SNNs) communicate through discrete spikes rather than continuous activations. The most widely used neuron model is the Leaky Integrate-and-Fire (LIF) neuron, whose membrane potential dynamics are described as:

$$H_t = \lambda V_{t-1} + C_t, \qquad S_t = \Theta(H_t - V_{th}), \qquad V_t = (1 - S_t) \cdot H_t + S_t \cdot V_{\text{reset}}, \qquad (1)$$

where $C_t$, $H_t$, $S_t$, and $V_t$ denote the input current, the accumulated membrane potential, the binary output spike, and the post-firing membrane potential at time step $t$, respectively. The parameters $V_{th}$, $V_{\text{reset}}$, and $\lambda$ represent the firing threshold, reset voltage, and leakage factor, respectively. $\Theta(\cdot)$ is the Heaviside step function.

## 3.2 REINFORCEMENT LEARNING

RL is a framework in which an agent learns to maximize cumulative rewards by interacting with an environment. The agent maps states (or observations) to actions, with the learning loop consisting of two steps: **(i)** the agent selects an action, receives a reward, and transitions to the next state; and **(ii)** the agent updates its policy by sampling mini-batches of past experiences.

Because the policy continuously evolves during training, the data distribution is inherently non-stationary. This poses challenges for batch normalization methods, which rely on the assumption of a stationary distribution.

## 3.3 BATCH NORMALIZATION

Batch Normalization (BN) (Ioffe & Szegedy, 2015) is a widely used technique to stabilize and accelerate the training of deep neural networks. Given an activation $x_i \in \mathbb{R}^d$ at iteration $i$, BN normalizes it using the mean and variance computed over a mini-batch $\mathcal{B} = \{x_i^1, \ldots, x_i^N\}$:

$$\mu_{\mathcal{B}} = \frac{1}{N} \sum_{j=1}^{N} x_i^j, \quad \sigma_{\mathcal{B}}^2 = \frac{1}{N} \sum_{j=1}^{N} (x_i^j - \mu_{\mathcal{B}})^2, \qquad (2)$$

$$\hat{x}_i = \frac{x_i - \mu_{\mathcal{B}}}{\sqrt{\sigma_{\mathcal{B}}^2 + \epsilon}}, \quad y_i = \gamma \hat{x}_i + \beta, \tag{3}$$

where $\epsilon$ is a small constant for numerical stability, and $\gamma, \beta$ are learnable affine parameters. During inference, moving statistics $(\hat{\mu}_i, \hat{\sigma}_i^2)$ are used in place of batch statistics $(\mu_i, \sigma_i^2)$.

In supervised learning, this discrepancy between training (mini-batch statistics) and inference (moving statistics) is usually tolerable, as imprecise moving estimates do not directly affect gradient updates. However, in online RL, inaccurate moving statistics degrade policy exploitation, leading to unstable training dynamics and even divergence.

## 4 METHODOLOGY

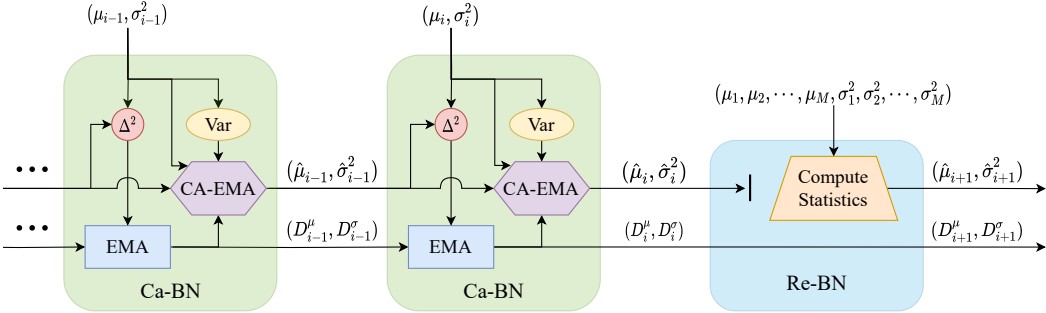

Figure 2: The statistics estimation scheme of CaRe-BN. In this framework, Ca-BN is applied at every update step, while Re-BN is performed periodically. $\Delta^2$ denotes the squared error, **Var** represents the variance computed according to Eq. 9, **EMA** refers to the exponential moving average in Eq. 11, and **CA-EMA** denotes the confidence-adaptive update defined in Eqs. 5 and 6.

As illustrated in Figure 2, we propose Confidence-adaptive and Recalibration Batch Normalization (CaRe-BN) to address the challenge of approximating moving statistics in online RL. Section 4.1 analyzes the limitations of traditional BN in online RL, where statistics are often estimated imprecisely. Section 4.2 introduces the confidence-adaptive update mechanism (Ca-BN), which dynamically adjusts statistics estimation based on the reliability of the current approximation. Section 4.3 presents the recalibration mechanism (Re-BN), which periodically corrects accumulated estimation errors. Finally, Section 4.4 integrates these components into the full CaRe-BN framework and demonstrates its use in online RL algorithms.

### 4.1 ISSUES IN APPROXIMATING MOVING STATISTICS

**Online RL introduces stronger distribution shifts.** Unlike supervised learning, where the data distribution is typically assumed to be static, online RL involves continuous interaction between the agent and the environment. This results in a non-stationary data distribution, which in turn causes activation statistics to drift over time.

**Inaccurate statistics degrade RL performance.** Supervised learning only requires the final moving statistics to be accurate, as inference is performed after training. In contrast, online RL requires reliable statistics throughout training. When statistics are imprecise, the agent selects suboptimal actions during exploration and exploitation, generating poor trajectories that further degrades policy updates.

The key of the problem lies in accurately estimating inference-time statistics under shifting distributions. Hence, it is essential to design estimators that adapt to distributional changes while minimizing approximation error during training.

It is worth noting that most conventional ANN-based RL algorithms do not employ BN (Lillicrap, 2015; Sutton & Barto, 2018), as shallow ANNs can often learn stable representations without normalization. In contrast, BN is indispensable for stabilizing SNNs training. Therefore, addressing this issue is particularly critical for SNN-based RL.

### 4.2 Confidence-adaptive Update of BN Statistics (Ca-BN)

Conventional BN approximates population statistics using an exponential moving average (EMA) of the batch mean and variance:

$$\hat{\mu}_i \leftarrow (1-\alpha)\hat{\mu}_{i-1} + \alpha\mu_i, \quad \hat{\sigma}_i^2 \leftarrow (1-\alpha)\hat{\sigma}_{i-1}^2 + \alpha\sigma_i^2, \tag{4}$$

where $\alpha$ is the momentum parameter. This update rule faces a fundamental **noise–delay trade-off**. As shown in Figure 1, low momentum yields stable but slow adaptation to distribution shifts, while high momentum adapts quickly but amplifies the noise from small-batch estimates. This trade-off is particularly harmful in online RL, where accurate normalization is critical for stable policy learning.

Inspired by the Kalman estimator (Kalman, 1960), we derive a confidence-guided mechanism that adaptively reweights estimators to minimize the mean-squared error (MSE) of BN statistics.

**Theorem 1** *Let $(\mu_i, \sigma_i^2)$ and $(\hat{\mu}_{i|i-1}, \hat{\sigma}_{i|i-1}^2)$ be two unbiased estimators of the population parameters $(\mu_i^*, \sigma_i^{*2})$. Taking them as random variables, the optimal linear estimator is*

$$\hat{\mu}_i = (1 - K_i^\mu)\hat{\mu}_{i|i-1} + K_i^\mu\mu_i, \qquad K_i^\mu = \frac{\mathbb{D}(\mu_i^* - \hat{\mu}_{i|i-1})}{\mathbb{D}(\mu_i^* - \hat{\mu}_{i|i-1}) + \mathbb{D}(\mu_i^* - \mu_i)}, \tag{5}$$

$$\hat{\sigma}_i^2 = (1 - K_i^\sigma)\hat{\sigma}_{i|i-1}^2 + K_i^\sigma\sigma_i^2, \qquad K_i^\sigma = \frac{\mathbb{D}(\sigma_i^{*2} - \hat{\sigma}_{i|i-1}^2)}{\mathbb{D}(\sigma_i^{*2} - \hat{\sigma}_{i|i-1}^2) + \mathbb{D}(\sigma_i^{*2} - \sigma_i^2)}, \tag{6}$$

*where $K_i^\mu$ and $K_i^\sigma$ are confidence-guided adaptive weights, and $\mathbb{D}(\cdot)$ denotes generalized variance[1].*

**Proof 1** *Since both $\hat{\mu}_{i|i-1}$ and $\mu_i$ are unbiased for $\mu_i^*$, any linear combination $\tilde{\mu}_i = (1-K)\hat{\mu}_{i|i-1} + K\mu_i$ is also unbiased. The variance is*

$$\mathbb{D}(\tilde{\mu}_i - \mu_i^*) = (1 - K)^2 \cdot \mathbb{D}(\hat{\mu}_{i|i-1} - \mu_i^*) + K^2 \cdot \mathbb{D}(\mu_i - \mu_i^*). \tag{7}$$

*Minimizing over $K$ yields the optimal $K = K_i^\mu$. The variance update (Eq. 6) follows analogously.*

**Assumption 1** *The activations in iteration $i$ are modeled as $x_i \sim \mathcal{N}(\mu_i^*, \sigma_i^{*2})$, following the standard Gaussianity assumption in BN.*

**Confidence of mini-batch statistics.** For a batch of size $N$, the sample mean $\mu_i$ and variance $\sigma_i^2$ satisfy

$$\mu_i \sim \mathcal{N}\left(\mu_i^*, \frac{\sigma_i^{*2}}{N}\right), \qquad \frac{(N-1)\sigma_i^2}{\sigma_i^{*2}} \sim \chi_{N-1}^2. \tag{8}$$

Since $\mu_i^*$ and $\sigma_i^{*2}$ are unknown, we adopt the common approximation using $\mu_i$ and $\sigma_i^2$, thus:

$$\mathbb{D}(\mu_i^* - \mu_i) = \frac{\sigma_i^{*2}}{N} \approx \frac{\sigma_i^2}{N}, \qquad \mathbb{D}(\sigma_i^{*2} - \sigma_i^2) = \frac{2\sigma_i^{*4}}{N-1} \approx \frac{2\sigma_i^4}{N-1}. \tag{9}$$

**Confidence of previous estimates.** Since the true statistics $\mu_i^*$ and $\sigma_i^{*2}$ are unknown, direct computation of $\mathbb{D}(\mu_i^* - \hat{\mu}_{i|i-1})$ and $\mathbb{D}(\sigma_i^{*2} - \hat{\sigma}_{i|i-1}^2)$ is infeasible. To approximate them, we view the minibatch statistics $\mu_i$ and $\sigma_i^2$ as a stochastic sample drawn from the unknown hypothetical distributions induced by $\mu_i^*$ and $\sigma_i^{*2}$. Thus, the squared deviations $(\mu_i - \hat{\mu}_{i|i-1})^2$ and $(\sigma_i^2 - \hat{\sigma}_{i|i-1}^2)^2$ serve as unbiased but noisy probes of $\mathbb{D}(\mu_i^* - \hat{\mu}_{i|i-1})$ and $\mathbb{D}(\sigma_i^{*2} - \hat{\sigma}_{i|i-1}^2)$.

Because these single-minibatch estimates exhibit high variance, we maintain smoothed recursive estimators updated using an exponential moving average with momentum parameter $\alpha$:

$$\mathbb{D}(\mu_i^* - \hat{\mu}_{i|i-1}) \approx D_i^\mu, \qquad \mathbb{D}(\sigma_i^{*2} - \hat{\sigma}_{i|i-1}^2) \approx D_i^\sigma, \tag{10}$$

$$D_i^\mu \leftarrow (1-\alpha)D_{i-1}^\mu + \alpha(\mu_i - \hat{\mu}_{i|i-1})^2, \qquad D_i^\sigma \leftarrow (1-\alpha)D_{i-1}^\sigma + \alpha(\sigma_i^2 - \hat{\sigma}_{i|i-1}^2)^2. \tag{11}$$

Combining Eqs. 5–11, we obtain the confidence-adaptive update scheme[2]. When distributional shifts are rapid, $D_i^\mu$ and $D_i^\sigma$ grow large, increasing $K_i^\mu$ and $K_i^\sigma$ and accelerating adaptation. Conversely, when statistics are stable, these terms shrink, lowering $K_i^\mu$ and $K_i^\sigma$ and reducing noise from small mini-batches.

---

[1]The confidence is defined as the inverse of the generalized variance: confidence score $= \frac{1}{\mathbb{D}}$.

[2]As BN statistics fluctuate without monotonic trends, we define $\hat{\mu}_{i|i-1} = \mu_{i-1}$ and $\hat{\sigma}_{i|i-1}^2 = \sigma_{i-1}^2$.

## 4.3 RE-CALIBRATION MECHANISM OF BN STATISTICS (RE-BN)

While the confidence-adaptive update provides online estimates of BN statistics during training, these estimates may still drift from the true population values due to stochastic mini-batch noise. The most accurate approach would be to recompute exact statistics by forward-propagating the entire dataset after each update (Wu & Johnson, 2021). However, this is computationally infeasible in RL, as it would require processing millions of samples at every step.

A more practical alternative is to periodically re-calibrate BN statistics using larger aggregated batches. Specifically, at fixed intervals $T_{\text{cal}}$, we draw $M$ calibration batches $\{\mathcal{B}_1, \ldots, \mathcal{B}_M\}$ from the replay buffer. For each batch $\mathcal{B}_j$, we compute its mean $\mu_j$ and variance $\sigma_j^2$. The recalibrated BN statistics are then given by:

$$\hat{\mu}_i = \frac{1}{M} \sum_{j=1}^{M} \mu_j, \qquad \hat{\sigma}_i^2 = \frac{1}{M} \sum_{j=1}^{M} (\sigma_j^2 + \mu_j^2) - \hat{\mu}_i^2. \tag{12}$$

This recalibration requires additional forward passes, but the extra overhead is upper bounded by $\frac{M}{T_{\text{cal}}}$ times the total training cost. Since we set $T_{\text{cal}} \gg M$, the computational overhead remains negligible, while significantly improving the accuracy of BN statistics.

## 4.4 INTEGRATING WITH RL

The proposed Confidence-adaptive and Re-calibration Batch Normalization (CaRe-BN) integrates two complementary mechanisms: the confidence-adaptive update in Section 4.2, which provides an online estimation of batch normalization (BN) statistics, and the re-calibration procedure in Section 4.3, which corrects accumulated bias. The overall integration within an online RL framework is outlined in Algorithm 1.

---

**Algorithm 1** General RL Algorithm with CaRe-BN

---

1: Initialize the agent networks and the replay buffer.
2: **for** each iteration **do**
3:     Select an action and store the transition (**inference BN statistics**).
4:     Update the agent by sampling a minibatch of $N$ transitions (**mini-batch BN statistics**).
5:     Update the moving BN statistics as:

$$D_i^{\mu} \leftarrow (1-\alpha)D_{i-1}^{\mu} + \alpha(\mu_i - \hat{\mu}_{i-1})^2, \qquad D_i^{\sigma} \leftarrow (1-\alpha)D_{i-1}^{\sigma} + \alpha(\sigma_i^2 - \hat{\sigma}_{i-1}^2)^2,$$

$$\hat{\mu}_i = \frac{D_i^{\mu} \cdot \mu_i + \frac{\sigma_i^2}{N} \cdot \hat{\mu}_{i-1}}{D_i^{\mu} + \frac{\sigma_i^2}{N}}, \qquad \hat{\sigma}_i^2 = \frac{D_i^{\sigma} \cdot \sigma_i^2 + \frac{2\sigma_i^4}{N-1} \cdot \hat{\sigma}_{i-1}^2}{D_i^{\sigma} + \frac{2\sigma_i^4}{N-1}}.$$

6:     **if** Re-calibration **then**
7:         Sample $M$ minibatches of $N$ transitions each and update BN statistics using Eq. (12).
8:     **end if**
9: **end for**

---

It is important to note that the inference procedure of CaRe-BN remains identical to that of conventional BN. At inference time, the CaRe-BN layer is seamlessly fused into synaptic weights, introducing no additional inference overhead during deployment.

## 5 EXPERIMENTS

### 5.1 EXPERIMENTAL SETUP

We evaluate CaRe-BN on RL tasks covering both discrete and continuous action spaces. All environments use default settings, and performance is evaluated by averaging the rewards in 10 trials.

For discrete action spaces, we consider four widely used Atari 2600 games from the Arcade Learning Environment (ALE) (Bellemare et al., 2013; Machado et al., 2018): *Pong*, *Breakout*, *SpaceInvaders*,

*Freeway*, and *Seaquest*. We adopt a deep Q-learning framework (Mnih et al., 2015) and train a deep Spiking Q-Network (Liu et al., 2022) that receives RAM-based observations and outputs state-action values.

For continuous control, we evaluate on five standard MuJoCo benchmarks (Todorov et al., 2012; Todorov, 2014b) provided in the OpenAI Gymnasium suite (Brockman, 2016; Towers et al., 2024): *InvertedDoublePendulum* (IDP) (Todorov, 2014a), *Ant* (Schulman et al., 2015), *HalfCheetah* (Wawrzyński, 2009), *Hopper* (Erez et al., 2012), and *Walker2d*. We employ a hybrid framework in which a spiking actor network is co-trained with a deep critic network using several RL algorithms, including Deep Deterministic Policy Gradient (DDPG) (Lillicrap, 2015), Twin Delayed DDPG (TD3) (Fujimoto et al., 2018), and Soft Actor-Critic (SAC) (Haarnoja et al., 2018a).

To evaluate the generality of CaRe-BN, we experiment with multiple spiking neuron models: the Leaky Integrate-and-Fire (LIF) neuron (Gerstner & Kistler, 2002), the Current-based LIF (CLIF) neuron (Tang et al., 2021), and the Dynamic Neuron (DN) model (Zhang et al., 2022), with detailed dynamics provided in the Appendix. All SNN agents are trained via Spatio-Temporal Backpropagation (STBP) (Wu et al., 2018), with the CaRe-BN module inserted between every pair of adjacent layers. For fair comparison, all models share the same hyperparameters, fully listed in the Appendix.

During each RL environment step, the SNN agent performs a single forward inference composed of 5 simulation time steps, after which all neuron states are reset.

## 5.2 More Precise BN Statistics Lead to Better Exploration

In online RL, the quality of exploration directly affects subsequent policy updates. As discussed in Section 4.1, traditional BN methods struggle to maintain accurate moving statistics, which can lead to suboptimal exploration behavior.

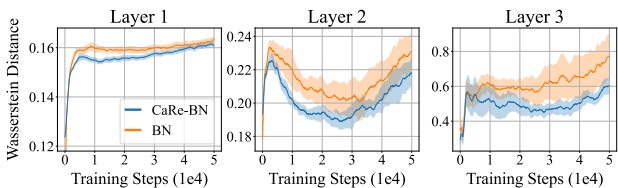

To quantify this effect, we compute the Wasserstein distance between the true feature distribution and the Gaussian distribution estimated by BN. Figure 3 shows that CaRe-BN consistently reduces this discrepancy across all layers throughout training, producing more precise normalization.

Figure 3: Wasserstein distance between estimated BN statistics and the true distribution across layers, measured with CLIF neurons and the TD3 algorithm in the InvertedDoublePendulum-v4 environment. Shaded areas denote half a standard deviation over five runs. Curves are uniformly smoothed for visual clarity.

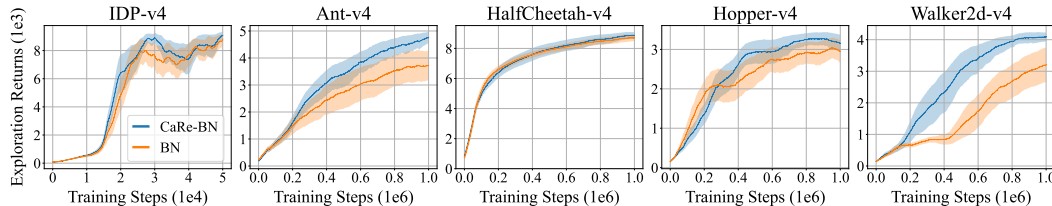

Figure 4: Exploration returns of BN and CaRe-BN with CLIF neurons and the TD3 algorithm across five MuJoCo tasks. Shaded areas represent half a standard deviation across five random seeds. Curves are uniformly smoothed for visual clarity.

The impact of improved statistics is reflected in exploration performance. As shown in Figure 4, CaRe-BN consistently achieves higher exploration returns. Since CaRe-BN does not directly modify the gradient update process, the observed improvement in exploration performance is solely due to its more precise estimation of BN statistics. This leads to better exploration policies, which in turn generate higher-quality trajectories for updating the agent. As a result, CaRe-BN forms a positive feedback loop: improved statistics → better exploration → higher-quality experiences → better policy.

## 5.3 ADAPTABILITY OF CARE-BN

To evaluate the adaptability of CaRe-BN, we test it across different RL algorithms (DQN (Mnih et al., 2015), DDPG (Lillicrap, 2015), TD3 (Fujimoto et al., 2018), and SAC[3] (Haarnoja et al., 2018a)) and spiking neuron models (LIF, CLIF (Tang et al., 2021), and DN (Zhang et al., 2022)).

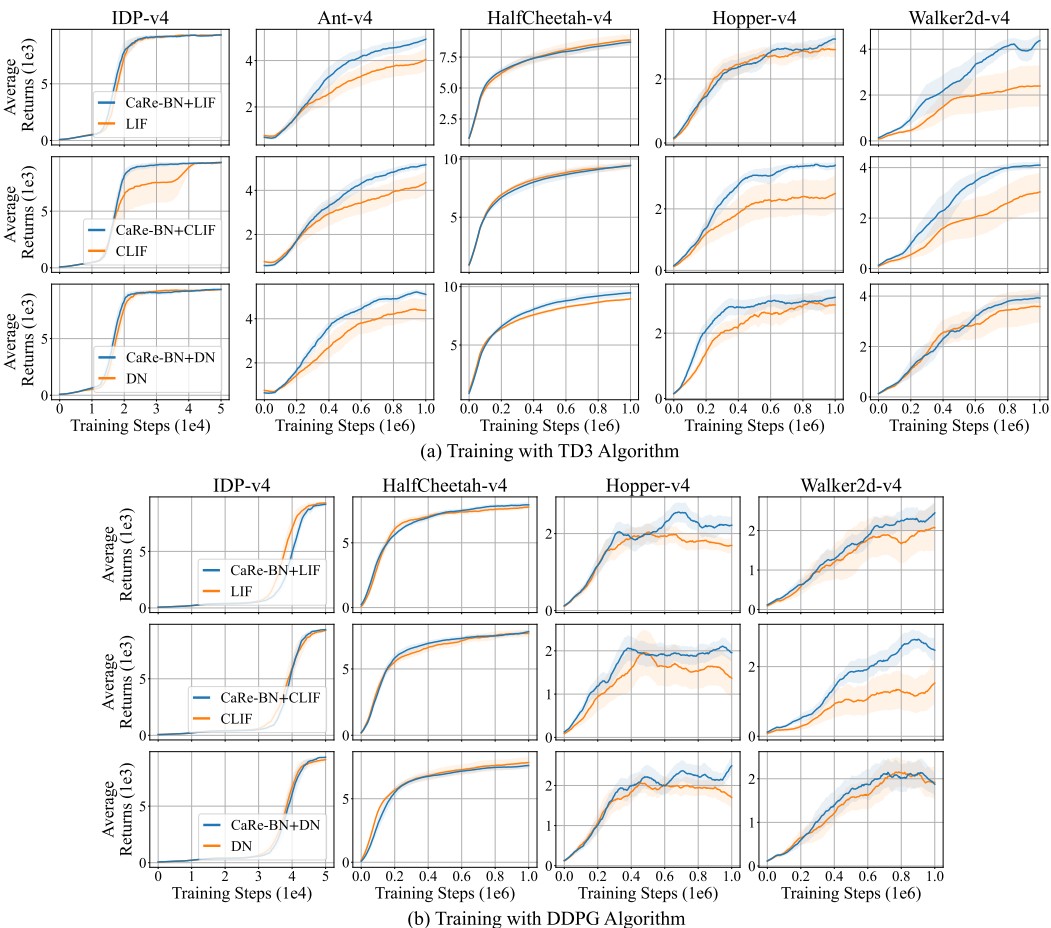

Figure 5: Learning curves of SNN-based agents in continuous control trained with TD3 (top) and DDPG (bottom). Since the DDPG algorithm (in both ANN and SNN) diverges in the Ant-v4 environment, these curves are not shown. Shaded areas represent half a standard deviation across five random seeds. Curves are uniformly smoothed for visual clarity.

**Better final return.** Figure 5 shows the learning curves for SNN models with and without CaRe-BN. In most cases, CaRe-BN consistently outperforms standard SNNs, converging faster and achieving higher final returns. These improvements are robust across different spiking neurons and RL algorithms, confirming that CaRe-BN enhances performance in diverse settings.

**Lower variance.** Figure 6 (a) and (b) display the relative variance of the final policy. Compared to standard SNNs, CaRe-BN significantly reduces the variance of SNN-RL training, and even achieves lower variance than ANN baselines (i.e., $17.71\%$ for DDPG and $21.24\%$ for TD3). This indicates that CaRe-BN not only enhances performance but also improves the stability and reproducibility.

**Generalizing across different RL domains.** Beyond continuous control, we also evaluate CaRe-BN in discrete-action settings using the deep spiking Q-network. As shown in Figure 7, SNN agents equipped with CaRe-BN achieve markedly improved performance across Atari tasks. These results demonstrate the strong generalization capability of CaRe-BN across diverse RL domains.

---

[3]Curves with SAC are shown in Figure 9 in the Appendix.

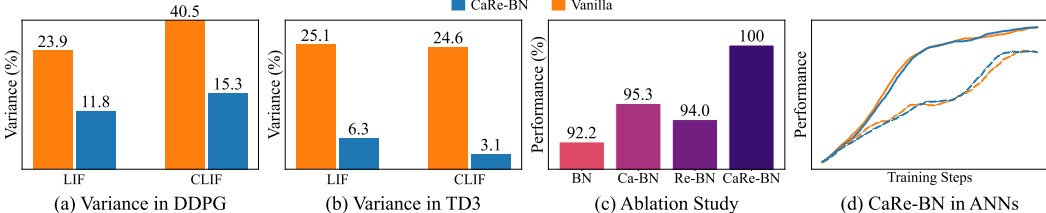

(a) Variance in DDPG  (b) Variance in TD3  (c) Ablation Study  (d) CaRe-BN in ANNs

Figure 6: (a), (b) Relative variance percentage of final policy returns, computed by averaging the standard deviation ratio across five random seeds, for all environments. (c) Normalized maximum performance across all environments for the ablation study, using CLIF neurons and TD3 algorithm. (d) Normalized learning curves across all environments for ANNs implementing CaRe-BN. The dashed lines represent DDPG and the solid lines represent TD3. Performance and training steps are normalized linearly. Curves are uniformly smoothed for visual clarity.

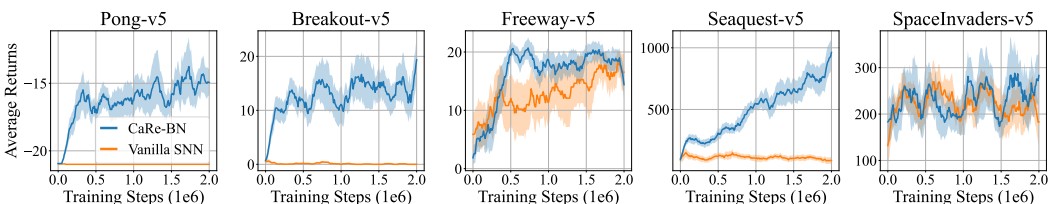

Figure 7: Learning curves of SNN-based agents in discrete control. Shaded areas represent half a standard deviation across three random seeds. Curves are uniformly smoothed for visual clarity.

## 5.4 EXCEEDING SOTA

To further validate the effectiveness of CaRe-BN, we compare it with existing state-of-the-art (SOTA) SNN-RL methods and various batch normalization strategies for SNNs. The evaluation is conducted using the TD3 algorithm (Fujimoto et al., 2018) (a strong SOTA baseline for continuous control) and the CLIF neuron model (Tang et al., 2021) (the most commonly used neuron type in recent SNN-RL studies). The ANN-SNN conversion baseline follows the SOTA method proposed in Bu et al. (2025). For direct-trained SNNs, we include pop-SAN (Tang et al., 2021), MDC-SAN (Zhang et al., 2022), and ILC-SAN (Chen et al., 2024a). Additionally, we test several BN algorithms for SNNs, including tdBN (Zheng et al., 2021), BNTT (Kim & Panda, 2021), TEBN (Duan et al., 2022), and TABN (Jiang et al., 2024). The performance is summarized in Table 1, where the average performance gain (APG) is defined as:

$$APG = \left( \frac{1}{|\text{envs}|} \sum_{\text{env} \in \text{envs}} \frac{\text{performance}(\text{env})}{\text{baseline}(\text{env})} - 1 \right) \cdot 100\%, \quad (13)$$

where $|\text{envs}|$ denotes the total number of environments, and performance(env) and baseline(env) represent the performance of the evaluated algorithm and the ANN baseline in each environment, respectively.

**Compared with other SNN-RL methods:** CaRe-BN significantly outperforms previous SNN-RL approaches, demonstrating that normalization plays a more crucial role than architectural modifications in improving SNN-RL performance.

**Compared with other BN methods:** Compared to existing SNN-specific BN variants, CaRe-BN performs superior, establishing a new state-of-the-art normalization strategy for SNN-RL.

**Compared with ANNs:** Notably, CaRe-BN trained with TD3 outperforms its ANN counterparts by $5.9\%$ on average[4]. This highlights that with proper normalization, SNNs can not only match but

---

[4] As shown in Figure 9 in the Appendix, SNNs equipped with CaRe-BN also outperform their ANN counterparts when trained with SAC (Haarnoja et al., 2018a).

Table 1: Max average returns over 5 random seeds with CLIF spiking neurons, and the average performance gain (APG) against ANN baseline, where $\pm$ denotes one standard deviation. All modules are trained using the TD3 algorithm. All directly trained SNN modules have 5 simulation time steps.

| Method | IDP-v4 | Ant-v4 | HalfCheetah-v4 | Hopper-v4 | Walker2d-v4 | APG |
|---|---|---|---|---|---|---|
| ANN | $7503 \pm 3713$ | $4770 \pm 1014$ | $10857 \pm 475$ | $3410 \pm 164$ | $4340 \pm 383$ | $0.00\%$ |
| ANN-SNN | $3859 \pm 4440$ | $3550 \pm 963$ | $8703 \pm 658$ | $3098 \pm 281$ | $4235 \pm 354$ | $-21.11\%$ |
| pop-SAN | $9351 \pm 1$ | $4590 \pm 1006$ | $9594 \pm 689$ | $2772 \pm 1263$ | $3307 \pm 1514$ | $-6.66\%$ |
| MDC-SAN | $9350 \pm 1$ | $4800 \pm 994$ | $9147 \pm 231$ | $3446 \pm 131$ | $3964 \pm 1353$ | $0.37\%$ |
| ILC-SAN | $9352 \pm 1$ | $5584 \pm 272$ | $9222 \pm 615$ | $3403 \pm 148$ | $4200 \pm 717$ | $4.64\%$ |
| tdBN | $9346 \pm 2$ | $4403 \pm 1134$ | $9402 \pm 527$ | $3592 \pm 46$ | $3464 \pm 970$ | $-2.28\%$ |
| BNTT | $9347 \pm 1$ | $4379 \pm 941$ | $9466 \pm 659$ | $3524 \pm 161$ | $3689 \pm 1247$ | $-1.62\%$ |
| TEBN | $9349 \pm 1$ | $4408 \pm 1156$ | $9452 \pm 539$ | $3472 \pm 135$ | $4235 \pm 381$ | $0.69\%$ |
| TABN | $9348 \pm 2$ | $4382 \pm 753$ | $9784 \pm 169$ | $3585 \pm 83$ | $4537 \pm 398$ | $3.25\%$ |
| CaRe-BN | $9348 \pm 2$ | $5373 \pm 159$ | $9563 \pm 442$ | $3586 \pm 49$ | $4296 \pm 268$ | $\mathbf{5.90}\%$ |

exceed the performance of traditional ANN-based RL agents, while retaining their energy-efficient advantages.

## 5.5 ABLATION STUDIES

We conduct ablation studies by separately evaluating the effects of the Confidence-adaptive update (Ca-BN) and the Re-calibration mechanism (Re-BN), as shown in Figure 6 (c). The results demonstrate that both the adaptive estimation and recalibration mechanisms are beneficial on their own. However, their combination provides the most significant improvement. Specifically, Ca-BN addresses the mismatch between training and inference statistics, while Re-BN corrects accumulated errors, further stabilizing training. By integrating both components, CaRe-BN achieves more precise and consistent normalization, leading to superior overall performance.

## 5.6 SNN-FRIENDLY DESIGN

Dispite the stunning improvement in SNNs, we also evaluate CaRe-BN on standard ANNs trained with TD3 and DDPG, as shown in Figure 6 (d). The results indicate that ANNs with CaRe-BN perform similarly to their baseline counterparts without CaRe-BN. This outcome is expected for the following reasons: (*i*) Shallow ANNs can already train stably and effectively without normalization[5], so adding CaRe-BN does not provide significant improvements. (*ii*) While CaRe-BN provides more precise estimates of BN statistics, this does not negatively impact the RL training process. These results further underscore that the improvements observed are not due to a stronger RL mechanism, but rather to the SNN-specific normalization strategies.

## 6 CONCLUSION

In this work, we introduced CaRe-BN, the first batch normalization method specifically designed for SNNs in RL. By addressing the instability of conventional BN in online RL, CaRe-BN enables SNNs to outperform their ANN counterparts in continuous control tasks. Importantly, CaRe-BN is lightweight and easy to integrate, making it a seamless drop-in replacement for existing SNN-RL pipelines without introducing additional computational overhead.

Beyond its technical contributions, CaRe-BN brings SNN-RL one step closer to practical deployment. By stabilizing training and improving exploration, it unlocks the potential of SNNs to act as both energy-efficient and high-performance agents in real-world continuous control applications. We believe this work underscores the importance of normalization strategies tailored to the unique dynamics of SNNs and opens new avenues for bridging the gap between neuromorphic learning and reinforcement learning at scale.

---

[5]In RL, networks typically consist of two hidden layers with 256 neurons.

ACKNOWLEDGMENTS

This work is supported by the National Natural Science Foundation of China (U24B20140, 62422601), Beijing Municipal Science and Technology Program (Z241100004224004), Beijing Nova Program (20230484362, 20240484703), National Key Laboratory for Multimedia Information Processing, and Beijing Key Laboratory of Brain-inspired Spiking Large Models.

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

## A    ETHICS STATEMENT

Our submission follows the ICLR Code of Ethics. We do not identify any specific ethical concerns in this work.

## B    REPRODUCIBILITY STATEMENT

Source code are provided in the supplementary materials. We also provide our full implementation and experimental configurations in the Appendix. All experiments were conducted on a single NVIDIA RTX 4090 GPU, but the code can also be executed on CPU-only devices, albeit with longer training times. These materials ensure that the reported results can be reproduced and verified by the community.

## C    USE OF LARGE LANGUAGE MODELS

Large Language Models (LLMs) were used solely for polishing the presentation of this paper, such as correcting typos, improving grammar. All ideas, derivations, algorithm design, and experiments were conceived and implemented independently **without** reliance on LLMs.

## D    APPENDIX

### D.1    SNN ARCHITECTURES

#### D.1.1    DEEP SPIKING Q-NETWORK ARCHITECTURE

The deep spiking Q-network consists of an SNN that receives the 128-dimensional RAM input using direct coding. The network contains two hidden layers, each with 256 LIF neurons. The Q-values are obtained by reading out the membrane potentials of the output layer, which uses non-leaky, non-firing neurons to provide stable value estimates.

#### D.1.2    SPIKING ACTOR NETWORK ARCHITECTURE

The spiking actor network (SAN) consists of a population encoder with Gaussian receptive fields, a multi-layer SNN with a population output, and a decoder with non-firing neurons.

**Forward Propagation of the SAN.**    In the state encoder, each input dimension is represented by $N_{\text{in}}$ soft-reset IF neurons with Gaussian receptive fields. These fields have trainable parameters $\mu$ and $\sigma$. The neurons receive stimulation $A_E$ at every time step and output spikes $S^{in}$ according to:

$$A_E = \exp\left[-\frac{1}{2}\frac{(s-\mu)^2}{\sigma^2}\right] \tag{14}$$

$$\begin{aligned} V_t^{in} &= V_{t-1}^{in} - S_{t-1}^{in} + A_E, \\ S_t^{in} &= \Theta(V_t^{in} - V_E), \end{aligned} \tag{15}$$

where $V_E$ is the threshold for the encoding populations.

The final layer of the SNN consists of $N_{\text{out}}$ neurons, corresponding to each action dimension. The decoder layer consists of non-spiking integrate-and-fire neurons connected to the last layer of the SNN:

$$V_t^{out} = V_{t-1}^{out} + W^{out} \cdot S_t^L + b^{out}, \tag{16}$$

where $W^{out}$ and $b^{out}$ are the weights and biases, respectively. The final output action is determined by the membrane potential at the last time step, $a = V_T^{out}$. A detailed description of the forward propagation in the spiking actor network is provided in Algorithm 2.

---

**Algorithm 2** Forward propagation of the Spiking Actor Network (SAN)

---
1: **Input:** $M_s$-dimensional observation $s$
2: Compute input population stimulation:

$$A_E = \exp\left[-\frac{1}{2}\frac{(s-\mu)^2}{\sigma^2}\right]$$

3: **for** $t = 1, \ldots, T$ **do**
4:      Compute encoder membrane potential and spikes:

$$V_t^{in} = V_{t-1}^{in} - S_{t-1}^{in} + A_E, \quad S_t^{in} = \Theta(V_t^{in} - V_E)$$

5:      **for** $l = 1, \ldots, L$ **do**
6:          Update neurons in layer $l$ at timestep $t$
7:      **end for**
8:      Update decoder membrane potential:

$$V_t^{out} = V_{t-1}^{out} + W^{out} \cdot S_t^L + b^{out}$$

9: **end for**
10: **Output:** $M_a$-dimensional action $a = V_T^{out}$

---

**Backpropagation of the SAN.** The SAN parameters are optimized using gradients with respect to the output action $a = V_T^{out}$, given $\frac{\partial L}{\partial a}$.

For the decoder:

$$\begin{aligned}
\frac{\partial L}{\partial W^{out}} &= \frac{\partial L}{\partial a} \cdot \frac{\partial V_T^{out}}{\partial W^{out}}, \\
\frac{\partial L}{\partial b^{out}} &= \frac{\partial L}{\partial a} \cdot \frac{\partial V_T^{out}}{\partial b^{out}}.
\end{aligned} \tag{17}$$

The main SNN is trained using spatio-temporal backpropagation (STBP) (Wu et al., 2018), with the rectangular surrogate gradient function defined as:

$$\Theta'(x) = \begin{cases} \frac{1}{2\omega}, & -\omega \leq x \leq \omega, \\ 0, & \text{otherwise}, \end{cases} \tag{18}$$

where $\omega$ denotes the window size.

Next, we derive the gradient of the encoder stimulation $A_E$, as shown in Eq. 19. For simplicity, the term $\frac{\partial S_t^{in}}{\partial A_E}$ is manually set to 1, which is a common surrogate assumption to simplify gradient computation:

$$\frac{\partial L}{\partial A_E} = \sum_{t=1}^{T} \frac{\partial L}{\partial S_t^{in}} \cdot \frac{\partial S_t^{in}}{\partial A_E} = \sum_{t=1}^{T} \frac{\partial L}{\partial S_t^{in}}. \tag{19}$$

Finally, the trainable parameters $\mu$ and $\sigma$ of the encoder can be updated as:

$$\begin{aligned}
\frac{\partial L}{\partial \mu} &= \frac{\partial L}{\partial A_E} \cdot \frac{\partial A_E}{\partial \mu} = \frac{\partial L}{\partial A_E} \cdot \frac{s-\mu}{\sigma^2} A_E, \\
\frac{\partial L}{\partial \sigma} &= \frac{\partial L}{\partial A_E} \cdot \frac{\partial A_E}{\partial \sigma} = \frac{\partial L}{\partial A_E} \cdot \frac{(s-\mu)^2}{\sigma^3} A_E.
\end{aligned} \tag{20}$$

### D.2 SPIKING NEURON MODELS

Section 3.1 introduced the LIF neuron model. Here, we provide the detailed dynamics of the spiking neuron models used in our experiments.

#### D.2.1 LIF NEURON MODEL

The dynamics of the LIF neuron are defined in Eq. 1, where the input current is computed as:

$$C_t^l = W^l S_t^{l-1} + b^l, \tag{21}$$

where $W$ and $b$ denote the synaptic weights and biases, respectively.

### D.2.2 CURRENT-BASED LIF (CLIF) NEURON MODEL

In the current-based LIF (CLIF) neuron proposed in Tang et al. (2021), the input current in Eq. 21 is modified as:

$$C_t^l = \lambda_c C_{t-1}^l + W^l S_t^{l-1} + b^l, \tag{22}$$

where $\lambda_c$ is the current leakage parameter. All other dynamics of CLIF neurons are identical to those of standard LIF neurons.

### D.2.3 DYNNMAIC NEURON MODEL

The second-order Dynamic Neuron (DN) model proposed in (Zhang et al., 2022) is designed to capture richer temporal dynamics for continuous control. Each DN maintains a membrane potential $V$ and a resistance variable $U$ to model hyperpolarization effects. The neuron dynamics are governed by:

$$\frac{dV_t^l}{dt} = (V_t^l)^2 - V_t^l - U_t^l + I_t^l, \tag{23}$$

$$\frac{dU_t^l}{dt} = \theta_v V_t^l - \theta_u U_t^l, \tag{24}$$

where $\theta_v$ and $\theta_u$ denote the conductance parameters of $V$ and $U$, respectively. When the neuron fires, the membrane potential $V$ is reset to $V_{\text{reset}}$, and the resistance variable $U$ is incremented by $\theta_s$. Using a first-order Taylor expansion, the iterative update of the DN model can be written as:

$$\begin{aligned}
C_t^l &= \alpha \cdot C_{t-1}^l + W^l S_t^{l-1} + b^l; \\
V_t^l &= \left(1 - S_{t-1}^l\right) \cdot V_{t-1}^l + S_{t-1}^l \cdot V_{\text{reset}}; \\
U_t^l &= U_{t-1}^l + S_{t-1}^l \cdot \theta_u; \\
V_{\text{delta}} &= V_t^{l^2} - V_t^l - U_t^l + C_t^l; \\
U_{\text{delta}} &= \theta_v \cdot V_t^l - \theta_u \cdot U_t^l; \\
V_t^l &= V_t^l + V_{\text{delta}}; \\
U_t^l &= U_t^l + U_{\text{delta}}; \\
S_t^l &= \Theta\left(V_t^l - V_{th}\right).
\end{aligned} \tag{25}$$

## D.3 EXPERIMENT DETAILS

### D.3.1 COMPUTE RESOURCES

All experiments were conducted on an RTX 4090 GPU (except for the training time study in Appendix D.5.1).

### D.3.2 SPIKING NEURON PARAMETERS

The parameters for the LIF and CLIF neurons are listed in Table 2. These are the same as those used in Tang et al. (2021), except that the LIF neuron does not include a current leakage parameter.

Table 2: Parameters of LIF and CLIF (Tang et al., 2021) neurons

| Parameter | LIF | CLIF (Tang et al., 2021) |
|---|---|---|
| Membrane leakage parameter $\lambda$ | 0.75 | 0.75 |
| Threshold voltage $V_{th}$ | 0.5 | 0.5 |
| Reset voltage $V_{\text{reset}}$ | 0 | 0 |
| Current leakage parameter $\alpha$ | - | 0.5 |

The parameters of the DN model are listed in Table 3. All values are obtained using the pre-learning procedure described in Zhang et al. (2022).

Table 3: Parameters of the DN (Zhang et al., 2022)

| Parameter | Value |
|---|---|
| SNN time steps | 5 |
| Threshold voltage $V_{th}$ | 0.5 |
| Current leakage parameter $\alpha$ | 0.5 |
| Conductivity of membrane potential $\theta_v$ | $-0.172$ |
| Conductivity of hidden state $\theta_u$ | 0.529 |
| Reset voltage $V_{\text{reset}}$ | 0.021 |
| spike effect to hidden state $\theta_s$ | 0.132 |

### D.3.3 Specific Parameters for CaRe-BN

Table 4 lists the hyperparameters of CaRe-BN. The recalibration frequency $T_{re}$ is set equal to the evaluation frequency used in the RL algorithms. All hyperparameters are kept consistent across different spiking neuron models and RL algorithms.

Table 4: Hyper-parameters of the CaRe-BN

| Parameter | Value |
|---|---|
| Momentum $\alpha$ | 0.8 |
| Recalibration frequency $T_{re}$ | 5000 |
| Recalibration batchs $M$ | 100 |

### D.3.4 Spiking Actor Network Parameters

All hyper-parameters of the spiking actor network are listed in Table 5. These settings are consistent with those used in a wide range of previous studies (Tang et al., 2021; Zhang et al., 2022; Chen et al., 2024a).

Table 5: Hyper-parameters of the spiking actor network

| Parameter | Value |
|---|---|
| Encoder population per dimension $N_{in}$ | 10 |
| Encoder threshold $V_E$ | 0.999 |
| Network hidden units | $(256, 256)$ |
| Decoder population per dimension $N_{out}$ | 10 |
| Surrogate gradient window size $\omega$ | 0.5 |

### D.3.5 RL Algorithm Parameters

The experiments are conducted using DQN (Mnih et al., 2015)), DDPG (Lillicrap, 2015), TD3 (Fujimoto et al., 2018), and the SAC (Haarnoja et al., 2018a) algorithms, with their respective hyperparameters listed in Tables 6, 7, 8, and 9.

### D.3.6 Experiment environments in continuous control

Figure 8 illustrates various MuJoCo environments (Todorov et al., 2012; Todorov, 2014b) from the OpenAI Gymnasium benchmarks (Brockman, 2016; Towers et al., 2024), including Inverted-DoublePendulum (IDP) (Todorov, 2014a), Ant (Schulman et al., 2015), HalfCheetah (Wawrzyński, 2009), Hopper (Erez et al., 2012), and Walker. All environments used the default configurations without modification.

Note that the state vectors, which can range from $-\infty$ to $\infty$, are normalized to $(-1, 1)$ using a tanh function. Similarly, since the actions have minimum and maximum limits, the outputs of the

Table 6: Hyper-parameters of the implemented DQN algorithm (Mnih et al., 2015)

| Parameter | Value |
|---|---|
| Learning rate | $1 \cdot 10^{-4}$ |
| Network architecture | $(256, 256)$ |
| Optimizer | Adam |
| Target update interval | 2000 |
| Batch size $N$ | 128 |
| Discount factor $\gamma$ | 0.99 |
| Iterations per time step | 1.0 |
| Reward scaling | 1.0 |
| Gradient clipping | None |
| Replay buffer size | $10^6$ |
| Max epsilon | 1 |
| Min epsilon | 0.1 |
| Epsilon decay steps | 20000 |

Table 7: Hyper-parameters of the implemented DDPG algorithm (Lillicrap, 2015)

| Parameter | Value |
|---|---|
| Actor learning rate | $1 \cdot 10^{-4}$ |
| Actor regularization | None |
| Critic learning rate | $1 \cdot 10^{-3}$ |
| Critic regularization | weight decay =0.01 |
| Critic architecture | $(400, 300)$ |
| Critic activation | Relu |
| Optimizer | Adam |
| Target update rate $\tau$ | $1 \cdot 10^{-3}$ |
| Batch size $N$ | 256 |
| Discount factor $\gamma$ | 0.99 |
| Iterations per time step | 1.0 |
| Reward scaling | 1.0 |
| Gradient clipping | None |
| Replay buffer size | $10^6$ |
| Exploration niose $\mathcal{N}(0, \sigma)$ | $\mathcal{N}(0, 0.1)$ |

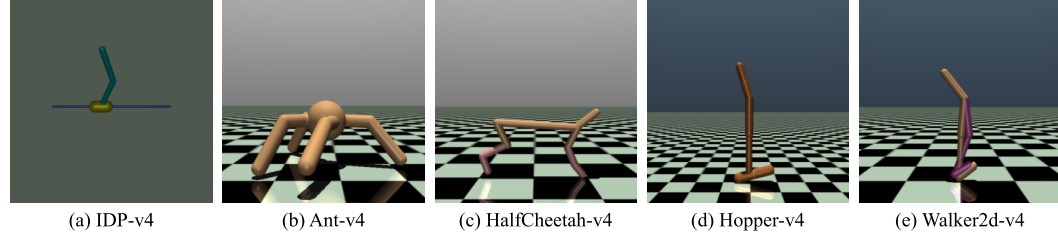

(a) IDP-v4     (b) Ant-v4     (c) HalfCheetah-v4     (d) Hopper-v4     (e) Walker2d-v4

Figure 8: Several continuous control tasks of the MuJoCo environments on OpenAI Gymnasium. (a) InvertedDoublePendulum-v4, (b) Ant-v4, (c) HalfCheetah-v4, (d) Hopper-v4, (e) Walker2d-v4.

Table 8: Hyper-parameters of the implemented TD3 algorithm (Fujimoto et al., 2018)

| Parameter | Value |
| --- | --- |
| Actor learning rate | $3 \cdot 10^{-4}$ |
| Actor regularization | None |
| Critic learning rate | $3 \cdot 10^{-4}$ |
| Critic regularization | None |
| Critic architecture | $(256, 256)$ |
| Critic activation | Relu |
| Optimizer | Adam |
| Target update rate $\tau$ | $5 \cdot 10^{-3}$ |
| Batch size $N$ | 256 |
| Discount factor $\gamma$ | 0.99 |
| Iterations per time step | 1.0 |
| Reward scaling | 1.0 |
| Gradient clipping | None |
| Replay buffer size | $10^6$ |
| Exploration niose $\mathcal{N}(0, \sigma)$ | $\mathcal{N}(0, 0.1)$ |
| Actor update interval $d$ | 2 |
| Target policy noise $\mathcal{N}(0, \tilde{\sigma})$ | $\mathcal{N}(0, 0.2)$ |
| Target policy noise clip $c$ | 0.5 |

Table 9: Hyper-parameters of the implemented SAC algorithm (Haarnoja et al., 2018a)

| Parameter | Value |
| --- | --- |
| Actor learning rate | $3 \cdot 10^{-4}$ |
| Actor regularization | None |
| Critic learning rate | $3 \cdot 10^{-4}$ |
| Critic regularization | None |
| Critic architecture | $(256, 256)$ |
| Critic activation | Relu |
| Optimizer | Adam |
| Target update rate $\tau$ | $10^{-3}$ |
| Batch size $N$ | 256 |
| Discount factor $\gamma$ | 0.99 |
| Iterations per time step | 1.0 |
| Reward scaling | 1.0 |
| Gradient clipping | None |
| Replay buffer size | $10^6$ |
| Actor update interval $d$ | 1 |
| Entropy target | $-dim(A)$ |
| Alpha learning rate | $3 \cdot 10^{-4}$ |

actor network are first normalized to $(-1, 1)$ via a tanh function and then linearly scaled to the corresponding (Min action, Max action) range.

## D.4 Additional Experimental Results

### D.4.1 Additional Results with SAC

In the main text, we demonstrated that CaRe-BN surpass its ANN counterparts using the TD3 algorithm. We further train the SNN agent using SAC, a stronger modern off-policy RL algorithm. As shown in Figure 9, SNNs equipped with CaRe-BN also have the potential to surpass their ANN counterparts under SAC.

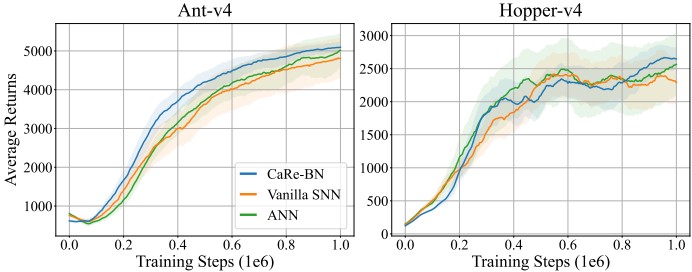

Figure 9: Learning curves of the SNN-based agents using SAC algorithm. Shaded areas represent half a standard deviation across five random seeds. Curves are uniformly smoothed for visual clarity.

### D.4.2 Additional Results on Adaptability

In the main text, we demonstrated that CaRe-BN improves performance across various spiking neuron models and RL algorithms. Additionally, Tables 10, 11, 12, 13, 14, and , 15 report the maximum average returns and the average performance gains of CaRe-BN compared to vanilla SNNs across different spiking neurons and RL algorithms.

Table 10: Max average returns over 5 random seeds in DDPG with LIF neurons.

| Method | IDP | HalfCheetah | Hopper | Walker2d | APG |
|---|---|---|---|---|---|
| Vanilla SNN | $9352 \pm 1$ | $7954 \pm 356$ | $3035 \pm 127$ | $2931 \pm 1395$ | **0.00%** |
| CaRe-BN | $9351 \pm 1$ | $8199 \pm 305$ | $3512 \pm 79$ | $3347 \pm 321$ | **8.24%** |

Table 11: Max average returns over 5 random seeds in DDPG with CLIF neurons.

| Method | IDP | HalfCheetah | Hopper | Walker2d | APG |
|---|---|---|---|---|---|
| Vanilla SNN | $9352 \pm 2$ | $8205 \pm 376$ | $2566 \pm 1270$ | $2224 \pm 1607$ | **0.00%** |
| CaRe-BN | $9352 \pm 0$ | $7972 \pm 245$ | $3247 \pm 100$ | $3709 \pm 321$ | **22.62%** |

Table 12: Max average returns over 5 random seeds in DDPG with DNs.

| Method | IDP | HalfCheetah | Hopper | Walker2d | APG |
|---|---|---|---|---|---|
| Vanilla SNN | $9351 \pm 3$ | $8069 \pm 897$ | $3134 \pm 134$ | $3238 \pm 633$ | **0.00%** |
| CaRe-BN | $9351 \pm 2$ | $7731 \pm 457$ | $3418 \pm 159$ | $3438 \pm 399$ | **2.76%** |

### D.4.3 Additional comparison with ANNs

Fig.10 shows the normalized learning curves of our CaRe-BN within different spiking neurons.

Table 13: Max average returns over 5 random seeds in TD3 with LIF neurons.

| Method | IDP | Ant | HalfCheetah | Hopper | Walker2d | APG |
|---|---|---|---|---|---|---|
| Vanilla SNN | $9347 \pm 1$ | $4243 \pm 949$ | $9073 \pm 946$ | $3507 \pm 85$ | $2807 \pm 1834$ | **0.00**% |
| CaRe-BN | $9346 \pm 1$ | $5083 \pm 356$ | $8813 \pm 533$ | $3489 \pm 118$ | $4556 \pm 497$ | **15.74**% |

Table 14: Max average returns over 5 random seeds in TD3 with CLIF neurons.

| Method | IDP | Ant | HalfCheetah | Hopper | Walker2d | APG |
|---|---|---|---|---|---|---|
| Vanilla SNN | $9351 \pm 1$ | $4590 \pm 1006$ | $9594 \pm 689$ | $2772 \pm 1263$ | $3307 \pm 1514$ | **0.00**% |
| CaRe-BN | $9348 \pm 2$ | $5373 \pm 159$ | $9563 \pm 442$ | $3586 \pm 49$ | $4296 \pm 268$ | **15.20**% |

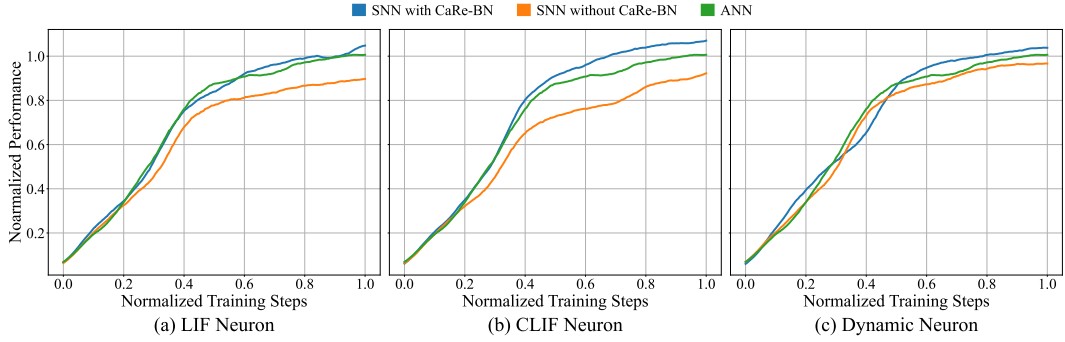

(a) LIF Neuron     (b) CLIF Neuron     (c) Dynamic Neuron

Figure 10: Normalized learning curves across all environments of the TD3 algorithm with different spiking neurons across all environments. The performance and training steps are normalized linearly based on ANN performance. Curves are uniformly smoothed for visual clarity.

#### D.4.4 ADDITIONAL COMPARISON WITH OTHER SNN-BN MECHANISMS

Tab. 16, shows the performance of different BN variants and CaRe-BN with the LIF neuron model in TD3 algorithm.

Table 16: Max average returns over 5 random seeds with LIF neuron, and the average performance gain (APG) against ANN baseline, where $\pm$ denotes one standard deviation.

| Method | IDP-v4 | Ant-v4 | HalfCheetah-v4 | Hopper-v4 | Walker2d-v4 | APG |
|---|---|---|---|---|---|---|
| ANN (TD3) | $7503 \pm 3713$ | $4770 \pm 1014$ | $10857 \pm 475$ | $3410 \pm 164$ | $4340 \pm 383$ | $0.00\%$ |
| Vanilla LIF | $9347 \pm 1$ | $4243 \pm 949$ | $9073 \pm 946$ | $3507 \pm 85$ | $2807 \pm 1834$ | $-7.08\%$ |
| tdBN | $9346 \pm 1$ | $4876 \pm 577$ | $8845 \pm 526$ | $3601 \pm 29$ | $4098 \pm 408$ | $1.65\%$ |
| BNTT | $9348 \pm 1$ | $5244 \pm 321$ | $9339 \pm 874$ | $3593 \pm 62$ | $3480 \pm 1450$ | $1.22\%$ |
| TEBN | $9347 \pm 1$ | $4408 \pm 1156$ | $9452 \pm 539$ | $3472 \pm 135$ | $4235 \pm 381$ | $0.69\%$ |
| TABN | $9347 \pm 1$ | $4431 \pm 1353$ | $9173 \pm 595$ | $3474 \pm 183$ | $3818 \pm 1133$ | $-1.64\%$ |
| CaRe-BN | $9346 \pm 1$ | $5083 \pm 356$ | $8813 \pm 533$ | $3489 \pm 118$ | $4556 \pm 497$ | $3.92\%$ |

Table 15: Max average returns over 5 random seeds in TD3 with DNs.

| Method | IDP | Ant | HalfCheetah | Hopper | Walker2d | APG |
|---|---|---|---|---|---|---|
| Vanilla SNN | $9350 \pm 1$ | $4800 \pm 994$ | $9147 \pm 231$ | $3446 \pm 131$ | $3964 \pm 1353$ | **0.00**% |
| CaRe-BN | $9349 \pm 2$ | $5444 \pm 161$ | $9581 \pm 638$ | $3470 \pm 115$ | $4084 \pm 362$ | **4.37**% |

### D.4.5 Additional results in ANN

We shows the normalized learning curves of the CaRe-BN with ANN in Fig.6 (d). Here, we show the detailed learning curves and maximum average returns of 5 environments in Fig.11, Fig.12, Tab.18 and Tab. 19, respectively.

### D.4.6 Additional results with different SNN simulation time steps.

We future study the impact of SNN simulation time steps. As shown in Table 17, SNNs generally benefit from larger simulation time steps, and CaRe-BN achieves even stronger results when using 8 SNN simulation steps (up to 6.32% improvement over ANNs). However, we report the main results using an SNN simulation time step of 5, following the standard configuration adopted in prior SNN-based RL studies (Tang et al., 2021; Zhang et al., 2022; Chen et al., 2024a).

Table 17: Max average returns over 5 random seeds of CaRe-BN with CLIF spiking neurons trained using the TD3 algorithm, and the average performance gain (APG) against ANN baseline, where $\pm$ denotes one standard deviation.

| SNN time steps | IDP-v4 | Ant-v4 | HalfCheetah-v4 | Hopper-v4 | Walker2d-v4 | APG |
|---|---|---|---|---|---|---|
| 2 | $953 \pm 247$ | $4924 \pm 171$ | $7635 \pm 392$ | $3588 \pm 10$ | $3885 \pm 1365$ | $-23.80\%$ |
| 3 | $9285 \pm 100$ | $5078 \pm 325$ | $8190 \pm 567$ | $3522 \pm 89$ | $4391 \pm 282$ | $2.03\%$ |
| 5 | $9348 \pm 2$ | $5373 \pm 159$ | $9563 \pm 442$ | $3586 \pm 49$ | $4296 \pm 268$ | $5.90\%$ |
| 8 | $9354 \pm 1$ | $5417 \pm 421$ | $9989 \pm 278$ | $3479 \pm 95$ | $4311 \pm 348$ | $6.32\%$ |

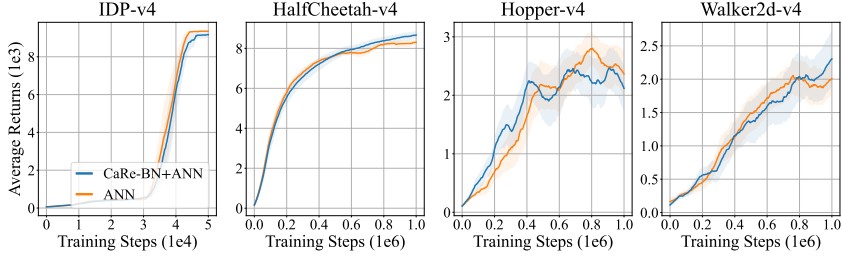

Figure 11: Learning curves of utilizing CaRe-BN in ANN with DDPG algorithm. The shaded region represents half a standard deviation over 5 different seeds. Curves are uniformly smoothed for visual clarity.

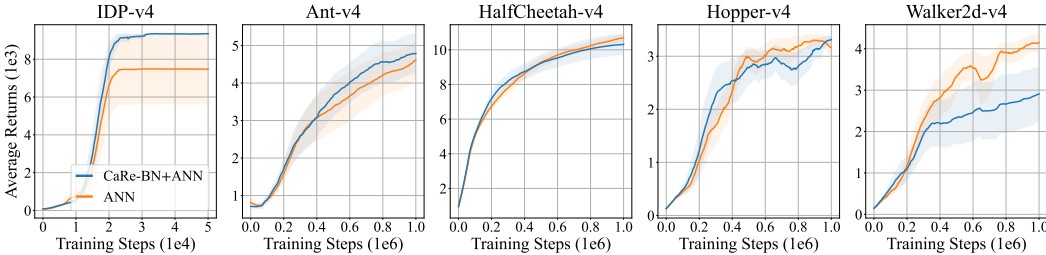

Figure 12: Learning curves of utilizing CaRe-BN in ANN with TD3 algorithm. The shaded region represents half a standard deviation over 5 different seeds. Curves are uniformly smoothed for visual clarity.

Table 18: Max average returns over 5 random seeds in DDPG with ANN.

| Method | IDP | HalfCheetah | Hopper | Walker2d | APG |
|---|---|---|---|---|---|
| Vanilla SNN | $9357 \pm 4$ | $8604 \pm 241$ | $3486 \pm 162$ | $3385 \pm 408$ | **0.00**% |
| CaRe-BN | $9360 \pm 0$ | $8887 \pm 408$ | $3475 \pm 135$ | $3328 \pm 882$ | **0.33**% |

Table 19: Max average returns over 5 random seeds in TD3 with ANN.

| Method | IDP | Ant | HalfCheetah | Hopper | Walker2d | APG |
|---|---|---|---|---|---|---|
| Vanilla SNN | $7503 \pm 3713$ | $4770 \pm 1014$ | $10857 \pm 475$ | $3410 \pm 164$ | $4340 \pm 383$ | **0.00**% |
| CaRe-BN | $9360 \pm 0$ | $5014 \pm 1122$ | $10458 \pm 1271$ | $3436 \pm 114$ | $3021 \pm 1360$ | **−0.69**% |

## D.5 ENERGY CONSUMPTIONS

### D.5.1 TRAINING COSTS

To assess the computational overhead introduced by CaRe-BN, we measure the training time and GPU memory usage on an RTX 3090 GPU paired with an Intel(R) Xeon(R) Platinum 8358P CPU. The results are summarized in Table 20. As shown, CaRe-BN does not introduce significant additional training time or memory consumption compared with other BN variants.

Table 20: Training costs of different BN mechanisms on the Ant-v4 environment, trained with TD3 algorithm and CLIF neurons. Training time corresponds to the total wall-clock time required for 5000 RL steps, including exploration, replay sampling, target computation, and gradient updates.

| Training costs | tdBN | BNTT | TEBN | TAB | CaRe-BN |
|---|---|---|---|---|---|
| Training time for 5000 updates (s) | 242 | 266 | 251 | 264 | 247 |
| GPU memory (MiB) | 437 | 437 | 441 | 441 | 437 |

### D.5.2 INFERRING COSTS

Table 21: Energy consumption per inference (in nJ) for the spiking actor network with CLIF neurons, trained using TD3 across various tasks.

| Method | IDP-v4 | Ant-v4 | HalfCheetah-v4 | Hopper-v4 | Walker2d-v4 | Average |
|---|---|---|---|---|---|---|
| ANN | 1715.20 | 1862.40 | 1785.60 | 1728.00 | 1785.60 | 1775.36 |
| SNN with CaRe-BN | 12.94 | 17.36 | 17.37 | 16.59 | 18.13 | 16.48 |

We evaluate the energy consumption of SNNs equipped with CaRe-BN during inference. Energy is estimated following the methodology of Merolla et al. (2014), where each floating-point operation (FLOP) is assumed to consume $12.5$ pJ and each synaptic operation (SOP) consumes $77$ fJ (Qiao et al., 2015; Hu et al., 2021). As shown in Table 21, the ANN baselines require substantially more energy per inference. In contrast, the SNN models with CaRe-BN demonstrate dramatically reduced energy consumption across all evaluated tasks. These results highlight the strong energy efficiency of SNNs and underscore their potential for deployment on resource-constrained platforms.

