# OpenReview forum: "CaRe-BN: Precise Moving Statistics for Stabilizing Spiking Neural Networks in Reinforcement Learning"
_ICLR.cc/2026/Conference — ICLR 2026 Poster_

### Official Review · Reviewer_eBqg · 2025-10-25

**Soundness:** 3
**Presentation:** 3
**Contribution:** 3
**Rating:** 6
**Confidence:** 5

**Summary:**

This work presents Confidence-adaptive and Re-calibration Batch Normalization (CaRe-BN) to stabilize Spiking Neural Networks (SNNs) in Reinforcement Learning (RL). Addressing imprecise BN statistics in online RL that hinder SNN performance, CaRe-BN integrates a confidence-guided adaptive update for BN statistics and a periodic recalibration mechanism, ensuring accurate normalization without disrupting RL training or compromising SNN energy efficiency at inference. Evaluated on MuJoCo continuous control tasks, it boosts SNN performance by up to 22.6% across neuron models/RL algorithms, even outperforming ANNs by 5.9%, advancing BN techniques for SNN-RL.

**Strengths:**

First, it addresses SNN-RL's BN statistic imprecision via dual mechanisms, stabilizing training without disrupting RL processes or losing SNN energy efficiency .

Second, it shows strong adaptability, boosting SNN performance by up to 22.6% across diverse neuron models and RL algorithms .

Third, it outperforms ANNs by 5.9%, proving SNNs’ potential for efficient, high-performance control .

**Weaknesses:**

First, it only evaluates on MuJoCo continuous control tasks, lacking tests on more complex or real-world RL scenarios .

Second, it fails to report comparisons of training time and memory with existing BN methods for SNN-RL .

Third, it doesn’t explore CaRe-BN’s performance with more advanced spiking neuron models.

**Questions:**

see Weaknesses

---

> ### Author Response · Authors · 2025-11-19
> **Reply 1/1**
>
> We are encouraged that you recognize the effectiveness of the proposed CaRe-BN in terms of adaptability and performance. We will do our best to address your concerns in detail.
>
> >W1: First, it only evaluates on MuJoCo continuous control tasks, lacking tests on more complex or real-world RL scenarios.
>
> Thank you for this valuable suggestion. Due to limited hardware resources, we are currently unable to test CaRe-BN in real-world RL applications. We acknowledge this as a limitation of this work, and we plan to explore real-world evaluations in future studies.
>
> In addition, we have added a new set of experiments on Atari 2600 games (Figure 7). SNN agents equipped with CaRe-BN achieve substantially better performance across these tasks. These results demonstrate that CaRe-BN not only improves SNN-RL in continuous control scenarios but is also effective in discrete action spaces. Based on these results, we believe CaRe-BN has the potential to generalize to more complex real-world settings.
>
>
> >W2: Second, it fails to report comparisons of training time and memory with existing BN methods for SNN-RL.
>
> We would first like to clarify that traditional RL algorithms typically **do not** employ BN, and existing BN methods for SNNs are not originally designed for online RL settings. Nonetheless, these SNN-specific BN variants can still be integrated into RL without modification, and we include them here for a comparison.
>
> All comparisons were conducted on a single RTX 3090 GPU with an Intel Xeon Platinum 8358P CPU. Training costs were measured on the Ant-v4 environment using TD3 and CLIF neurons. Training time is reported as the total wall-clock time required for 5000 RL updates, including exploration, replay sampling, target computation, and gradient updates.
>
>
> **Table R1:** Training time and GPU memory comparison for different BN variants.
> |Training costs|tdBN|BNTT|TEBN|TAB|**CaRe-BN**|
> | -------- | :--------: | :--------: | :--------: | :--------: | :--------: |
> |Training time for 5000 updates (s)|242|266|251|264|247|
> |GPU memory (MiB)|437|437|441|441|437|
>
> As shown in Table R1, CaRe-BN does not introduce any noticeable additional training time or memory usage compared to existing BN variants, demonstrating that our method incurs minimal computational overhead.
>
> We have added these results and their discussion in Table 20 and Appendix D.5.1 in the revised version.
>
>
> >W3: Third, it doesn’t explore CaRe-BN’s performance with more advanced spiking neuron models.
>
> Thank you for this valuable suggestion.
>
> To further validate the generality of CaRe-BN, we added experiments using the second-order dynamic neuron model (DN), which has been shown to be effective in RL settings [1]. The DN consists of a membrane potential $V$ and a resistance item $U$ to simulate hyperpolarization. Its dynamics are given as follows:
> $$\frac{d V_t^l}{d t}={V_t^l}^{2}-V_t^l-U_t^l+I_t^l,$$$$\frac{d U_t^l}{d t}=\theta_{v} V_t^l-\theta_{u} U_t^l,
> $$where $\theta_{v}$ and $\theta_{U}$ are the conductivities of $V$ and $U$, respectively. We follow the neuron hyperparameters recommended in [1].
>
> We have added these results to Figure 5 in the revised manuscript. Across multiple environments and RL algorithms, SNNs equipped with dynamic neurons also benefit from CaRe-BN. These results further demonstrate the robustness and generalizability of CaRe-BN across different neuron models.
>
>
> **References:**
>
> [1] Duzhen Zhang, Tielin Zhang, Shuncheng Jia, and Bo Xu. Multi-scale dynamic coding improved spiking actor network for reinforcement learning. In Proceedings of the AAAI conference on artificial intelligence, volume 36, pp. 59–67, 2022.

---

> > ### Comment · Reviewer_eBqg · 2025-11-22
> >
> > The authors have supplemented new experiments that effectively demonstrate the generalization ability, maintained performance, and applicability of the proposed method to complex tasks. I appreciate the authors' comprehensive responses to my previous concerns, which have now been fully addressed. Accordingly, I am willing to raise my evaluation score to express my support for this work.

---

> > > ### Author Response · Authors · 2025-11-22
> > > **Thank You**
> > >
> > > We sincerely appreciate your thoughtful review. Thank you for recognizing our work and raising the score.

---

### Official Review · Reviewer_EiPv · 2025-10-29

**Soundness:** 2
**Presentation:** 3
**Contribution:** 2
**Rating:** 6
**Confidence:** 3

**Summary:**

This paper highlights the misalignment between the true and computed activation distributions when batch normalization is applied in spiking neural networks (SNNs). In particular, the work discusses issues with existing batch normalization methods in SNN models that can negatively impact performance in reinforcement learning (RL) tasks.

Inspired by the Kalman filter, the authors propose a confidence-guided approach that more accurately estimates batch statistics. They also discuss a recalibration method to align the computed batch statistics with those of the training data.

The proposed method is evaluated on several RL tasks and compared against multiple baselines from both ANN- and SNN-based RL frameworks. Furthermore, an ablation study is conducted to demonstrate the performance improvements contributed by each individual component of the method.

**Strengths:**

Most of the paper is easy to follow.

The problem highlighted is an interesting issue in BN for reinforcement learning.

I believe the discussion of related work is adequate and also provides the necessary background to understand the problem.

**Weaknesses:**

I believe the novel part of the method is the confidence adaptive updation of batch statistics; however, the second part, recalibration, is a modification of the existing method.

A few things are not clear, which are mostly related to the proposed method, which can be found in the questions section.

**Questions:**

Line 204: The authors mention that “conventional ANN-based RL algorithms do not employ BN.” This raises an important question — is batch normalization (BN) truly necessary in SNN-based RL frameworks? How does the proposed SNN model perform without BN? A comparison or ablation along these lines would strengthen the justification for including BN in SNNs.


Line 233: The authors provide a proof for computing the optimal K value. It appears that \mathbb{D} is computed recursively. Could the authors please elaborate on this derivation and clarify the recursive relationship involved?


The definition and interpretation of the measure \mathbb{D} are unclear. Could the authors provide more explanation regarding what \mathbb{D}  quantifies and how it relates to the overall estimation or confidence process?


The rationale behind approximating \mathbb{D} as shown in Equation (9) is not clearly justified. Please clarify why this specific approximation is valid and under what assumptions it holds.


The terminology “confidence-based adaptive scheme” is somewhat confusing. Why is the deviation from the actual value interpreted as confidence? A clearer explanation of how “confidence” is defined and why it is used in this context would be helpful.

---

> ### Author Response · Authors · 2025-11-19
> **Reply 1/3**
>
> We are encouraged that you think the issue is interesting. We will strive to address your questions.
>
> >W1: I believe the novel part of the method is the confidence adaptive updation of batch statistics; however, the second part, recalibration, is a modification of the existing method.
>
>
> Thank you for the thoughtful comment. We agree that the primary novelty of the CaRe-BN is the confidence adaptive update of BN statistics (Ca-BN), while the re-calibration (Re-BN) is a comparatively smaller contribution.
>
> However, we would like to emphasize that the novelty of Re-BN does not lie in reusing existing recalibration ideas, but rather in designing a principled and low-cost way to obtain more precise BN statistics under the unique constraints of online RL.
>
>
> >Q1: Line 204: The authors mention that “conventional ANN-based RL algorithms do not employ BN.” This raises an important question — is batch normalization (BN) truly necessary in SNN-based RL frameworks? How does the proposed SNN model perform without BN? A comparison or ablation along these lines would strengthen the justification for including BN in SNNs.
>
> Thank you for this insightful question. We guess there might be a misunderstanding. We would like to clarify that all comparisons presented in Figure 5 and Figure 7 directly evaluate **SNNs without BN** versus **SNNs equipped with CaRe-BN**. Moreover, existing SNN-based RL methods do not employ BN, meaning these experiments already reflect the current standard practice.
>
> Theoretically, SNNs rely heavily on BN due to their inherently unstable gradient propagation, including vanishing and exploding gradients, which is caused by the discrete spike and the surrogate gradient function. This problem is shared across both supervised learning and RL in SNNs.
>
> Empirically, as shown in Figures 5 and 7, SNNs without BN (traditional approaches) consistently perform poorer than SNNs with CaRe-BN. These results demonstrate that BN is essential for stable and effective SNN-RL training. In addition, as discussed in Section 5.6, ANNs do not benefit from CaRe-BN, while SNNs improve substantially, further highlighting that SNNs particularly require proper BN in the field of RL.
>
> We have updated the Abstract and Introduction to more explicitly highlight the necessity of BN in SNN-based RL frameworks.

---

> ### Author Response · Authors · 2025-11-19
> **Reply 2/3**
>
> >Q2: Line 233: The authors provide a proof for computing the optimal K value. It appears that $\mathbb{D}$ is computed recursively. Could the authors please elaborate on this derivation and clarify the recursive relationship involved?
>
> We think you are referring to the recursive estimation of $\mathbb{D}(\mu _ i ^ * - \hat{\mu} _ {i\mid i-1})$ and $\mathbb{D}({\sigma _ i ^ *} ^ 2 - \hat{\sigma} ^ 2 _ {i\mid i-1})$, which are computed as:
> $$
> \mathbb{D}(\mu _ i ^ * - \hat{\mu} _ {i\mid i-1}) \approx D _ i ^ \mu,
> \qquad
> D ^ \mu _ i \gets (1-\alpha)D ^ \mu _ {i-1} + \alpha(\mu _ i-\hat\mu _ {i-1}) ^ 2,
> $$$$
> \mathbb{D}({\sigma _ i ^ *} ^ 2 - \hat{\sigma} ^ 2 _ {i\mid i-1}) \approx D _ i ^ \sigma,
> \qquad
> D ^ \sigma _ i \gets (1-\alpha)D ^ \sigma _ {i-1} + \alpha(\sigma _ i ^ 2-\hat\sigma ^ 2 _ {i-1}) ^ 2,
> $$where $\mathbb{D}$ denotes the variance. $\mu _ i ^ *$ and ${\sigma _ i ^ *} ^ 2$ are fixed but unknown target statistics, we treat them as random variables from unknown hypothetical distributions. The estimators $\hat{\mu} _ {i\mid i-1}$ and $\hat{\sigma} ^ 2 _ {i\mid i-1}$ are computed from the previous iteration’s statistics:
> $$\hat{\mu} _ {i\mid i-1}=\mu _ {i-1}, \qquad\hat{\sigma} ^ 2 _ {i\mid i-1}=\sigma ^ 2 _ {i-1},$$ as we assume BN statistics fluctuate without any monotonic trend.
>
> Thus, we can formally get:
> $$\mathbb{D}(\mu _ i ^ * - \hat{\mu} _ {i\mid i-1})=\mathbb{E} _ {\tilde\mu _ i}(\tilde\mu _ i - \mu _ {i-1}) ^ 2,
> \qquad
> \mathbb{D}({\sigma _ i ^ *} ^ 2 - \hat{\sigma} ^ 2 _ {i\mid i-1})=\mathbb{E} _ {\tilde\sigma ^ 2}(\tilde\sigma ^ 2 - \sigma ^ 2 _ {i-1}) ^ 2,
> $$where $\tilde\mu _ i$ and $\tilde\sigma ^ 2$ are hypothetical samples from the unknown hypothetical distributions.
>
> In practice, each minibatch sampled from the replay buffer is treated as one sample from these hypothetical distributions. Thus, we can derive:
> $$\mathbb{D}(\mu _ i ^ * - \hat{\mu} _ {i\mid i-1})\approx(\mu _ i - \hat{\mu} _ {i-1}) ^ 2,
> \qquad
> \mathbb{D}({\sigma _ i ^ *} ^ 2 - \hat{\sigma} ^ 2 _ {i\mid i-1})\approx(\sigma _ i ^ 2 - \hat{\sigma} ^ 2 _ {i-1}) ^ 2.
> $$However, these single-sample estimates are very noisy, so we apply an exponential moving average (EMA), which is the recursive form shown in the manuscript:
> $$
> \mathbb{D}(\mu _ i ^ * - \hat{\mu} _ {i\mid i-1}) \approx D _ i ^ \mu,
> \qquad
> D ^ \mu _ i \gets (1-\alpha)D ^ \mu _ {i-1} + \alpha(\mu _ i-\hat\mu _ {i-1}) ^ 2,
> $$$$
> \mathbb{D}({\sigma _ i ^ *} ^ 2 - \hat{\sigma} ^ 2 _ {i\mid i-1}) \approx D _ i ^ \sigma,
> \qquad
> D ^ \sigma _ i \gets (1-\alpha)D ^ \sigma _ {i-1} + \alpha(\sigma _ i ^ 2-\hat\sigma ^ 2 _ {i-1}) ^ 2.$$
>
> We have revised Section 4.2 in the manuscript to make this recursive derivation clearer.
>
>
> >Q3: The definition and interpretation of the measure $\mathbb{D}$ are unclear. Could the authors provide more explanation regarding what $\mathbb{D}$ quantifies and how it relates to the overall estimation or confidence process?
>
> Thank you for this question. In our work, $\mathbb{D}$ denotes the variance of a random variable drawn from a distribution (can be hypothetical), which serves as a measure of uncertainty.
>
> For $\mathbb{D}(\mu _ i ^ *-\mu _ i)$ and $\mathbb{D}({\sigma _ i ^ *} ^ 2-\sigma _ i ^ 2)$, the mini-batch statistics $\mu _ i$ and $\sigma _ i ^ 2$ are treated as random variables due to sampling randomness.
>
> For $\mathbb{D}(\mu _ i ^ * - \hat{\mu} _ {i\mid i-1})$ and $\mathbb{D}(\mu _ i ^ * - \hat{\mu} _ {i\mid i-1})$, the unknown target statistics $\mu _ i ^ *$ and ${\sigma _ i ^ *} ^ 2$ are viewed as random variables sampled from the hypothetical distributions.
>
> Specifically, we use $\tilde\mu _ i$ and $\tilde\sigma ^ 2$ to denote samples drawn from the distributions, the mesaurments of $\mathbb{D}$ are defined as:$$
> \mathbb{D}(\mu _ i ^ *-\mu _ i)=\mathbb{E} _ {\tilde\mu _ i}(\mu _ i ^ *-\tilde\mu _ i) ^ 2,
> \qquad
> \mathbb{D}({\sigma _ i ^ *} ^ 2-\sigma _ i ^ 2)=\mathbb{E} _ {\tilde\sigma ^ 2}({\sigma _ i ^ *} ^ 2-\tilde\sigma ^ 2) ^ 2,
> $$$$
> \mathbb{D}(\mu _ i ^ * - \hat{\mu} _ {i\mid i-1})=\mathbb{E} _ {\tilde\mu _ i}(\tilde\mu _ i - \hat{\mu} _ {i\mid i-1}) ^ 2,
> \qquad
> \mathbb{D}({\sigma _ i ^ *} ^ 2 - \hat{\sigma} ^ 2 _ {i\mid i-1})=\mathbb{E} _ {\tilde\sigma ^ 2}(\tilde\sigma ^ 2 - \hat{\sigma} ^ 2 _ {i\mid i-1}) ^ 2,
> $$where $\mathbb{E}$ denotes the expectation.
>
>
> Regarding its relation to the overall estimation and confidence: The variance mesaurments $\mathbb{D}$ quantify the uncertainty, which is the inverse of the confidence value, as the larger the variance, the lower the confidence.

---

> ### Author Response · Authors · 2025-11-19
> **Reply 3/3**
>
> >Q4: The rationale behind approximating $\mathbb{D}$ as shown in Equation (9) is not clearly justified. Please clarify why this specific approximation is valid and under what assumptions it holds.
>
> Under Assumption 1, for a mini-batch of size $N$, the sample mean $\mu_i$ and sample variance $\sigma_i^2$ follow the distributions:
> $$
>     \mu _ i \sim \mathcal{N}\left(\mu _ i ^ *, \tfrac{{\sigma _ i ^ *} ^ 2}{N}\right),
> \qquad
> \frac{(N-1)\sigma _ i ^ 2}{{\sigma _ i ^ *} ^ 2}\sim \chi ^ 2 _ {N-1}.
> $$From these distributions, $\mathbb{D}$ can be computed analytically by:
> $$
> \mathbb{D}(\mu _ i ^ *-\mu _ i)=\text{Variance}\left(\mathcal{N}\left(\mu _ i ^ *, \tfrac{{\sigma _ i ^ *} ^ 2}{N}\right)\right)=\frac{{\sigma _ i ^ *} ^ 2}{N}
> $$$$
> \mathbb{D}({\sigma_i ^ *} ^ 2-\sigma _ i ^ 2)=\frac{{\sigma _ i ^ *} ^ 4}{(N-1)^2}\cdot\text{Variance}\left(\chi^2 _ {N-1}\right)=\frac{2{\sigma _ i ^ *} ^ 4}{N-1}
> $$However, the true statistic ${\sigma _ i ^ *} ^ 2$ is unknown in practice. Therefore, we approximate it using the observed mini-batch statistic ${\sigma _ i} ^ 2$. This yields the approximation used in Equation (9).
>
> While this substitution introduces noise, the approximation remains reliable because the mini-batch size is sufficiently large (typically 128–256 in continuous control). The resulting error appears only as a first-order noise term in $\mathbb{D}(\mu _ i ^ *-\mu _ i)$ and $\mathbb{D}({\sigma _ i ^ *} ^ 2-\sigma _ i ^ 2)$, and subsequently as a first-order noise term in $K^\mu_i$ and $K^\sigma _ i$. Since CaRe-BN adapts $K ^ \mu _ i$ and $K ^ \sigma _ i$ at the zero-order level, this level of noise is tolerable.
>
>
> >Q5: The terminology “confidence-based adaptive scheme” is somewhat confusing. Why is the deviation from the actual value interpreted as confidence? A clearer explanation of how “confidence” is defined and why it is used in this context would be helpful.
>
> Thank you for this question. We have revised the manuscript to clarify this.
>
> We define the confidence of an estimator as the reciprocal of its estimation variance,
> $$\text{Conf}=\frac{1}{\mathbb{D}},$$
> which is conventional in statistics and filtering, as higher variance indicates lower confidence.
>
> Based on these definitions, the optimal linear estimators in Eq.(5) and Eq.(6) can be written as:
> $$
> \hat\sigma_i^2 = \frac{\text{Conf}(\hat\sigma^2_{i\mid i-1})\cdot\hat\sigma^2_{i\mid i-1}+\text{Conf}(\sigma_i^2)\cdot\sigma_i^2}{\text{Conf}(\hat\sigma^2_{i\mid i-1})+\text{Conf}(\sigma_i^2)} ,
> \qquad
> \hat\mu_i = \frac{\text{Conf}(\hat\mu_{i\mid i-1})\cdot\hat\mu_{i\mid i-1}+\text{Conf}(\mu_i)\cdot\mu_i}{\text{Conf}(\hat\mu_{i\mid i-1})+\text{Conf}(\mu_i)}.
> $$

---

> > ### Comment · Reviewer_EiPv · 2025-11-26
> > **Thanks for the detailed response**
> >
> > I would like to thank the authors for their detailed response and for updating the manuscript accordingly. For this reason, I am increasing my score.

---

> > > ### Author Response · Authors · 2025-11-26
> > > **Thank You**
> > >
> > > We sincerely appreciate your recognition of our work. Thank you for the constructive review and the improved score.

---

### Official Review · Reviewer_Yg5y · 2025-10-30

**Soundness:** 2
**Presentation:** 3
**Contribution:** 2
**Rating:** 2
**Confidence:** 3

**Summary:**

The paper proposes a method to improve the estimation of Batch Normalization (BN) statistics. The authors apply this improved BN to spiking neural networks (SNNs) and report that it outperforms other BN variants on six reinforcement learning (RL) tasks.

**Strengths:**

1. The paper is relatively easy to follow.

2. Multiple tasks are evaluated.

**Weaknesses:**

1. The claim that “due to the discrete and non-differentiable nature of spikes, directly trained SNNs rely heavily on BN to stabilize gradient updates” is misleading. BN is not designed to solve non-differentiability in SNNs; please revise this.

2. It is unclear whether the contribution is primarily for ANNs or SNNs. Figure 1 suggests general applicability to ANNs, yet experiments are focusing on SNNs—please clarify the intended scope and update the paper.

3. Section 5.2 / Figure 3: The evidence in Figure 3 appears insufficient to conclude that the method yields more precise BN statistics, and contributes to better reward. We can only observe that CaRe-BN gets obviously higher expected rewards on three tasks out of five. It is unknown what is the root cause of such observation.

4. How many time steps are used in the SNNs? How sensitive is performance to this hyperparameter, and what justifies the chosen value to be used in the baseline setting? Also, ablation study on the time step should be provided.

**Questions:**

1. Figure 1: What does the “update step” represent—training parameter updates or an SNN inference time step?

2. I could not find details of the network architectures used. Please describe the models (layers, widths, neuron types, readouts, etc.).

3. How are SNN time steps integrated with the RL policy? Do you aggregate/compress time steps to produce the actor/critic outputs? Which components are SNNs and which components are not?

---

> ### Author Response · Authors · 2025-11-19
> **Reply 1/3**
>
> We would like to thank you for your constructive questions and suggestions. We will conduct a detailed clarification and discussion for your concerns.
>
> >W1: The claim that “due to the discrete and non-differentiable nature of spikes, directly trained SNNs rely heavily on BN to stabilize gradient updates” is misleading. BN is not designed to solve non-differentiability in SNNs; please revise this.
>
> Thank you for pointing this out. We apologize that our original wording was inappropriate and might cause misunderstanding.
>
> We have revised the sentence in the Abstract to:
> ```
> However, the discrete and non-differentiable nature of spikes leads to unstable gradient propagation in directly trained SNNs, making Batch Normalization (BN) an important component for stabilizing training.
> ```
> We have revised the sentences in the Introduction to:
> ```
> Due to the discrete spike dynamics and the reliance on surrogate gradients to approximate the backward pass, directly trained SNNs often suffer from unstable gradient propagation, including vanishing or exploding gradients. Batch Normalization (BN) plays a crucial role in stabilizing SNN training by regulating activation statistics and improving gradient flow, mitigates such instability and contributes to state-of-the-art performance.
> ```
>
>
>
>
> >W2: It is unclear whether the contribution is primarily for ANNs or SNNs. Figure 1 suggests general applicability to ANNs, yet experiments are focusing on SNNs—please clarify the intended scope and update the paper.
>
>
> Thank you for this question. We would like to clarify that the contribution of this work is **specifically targeted at SNN-based RL**.
>
> Our motivation arises from a key challenge: while traditional RL algorithms for ANNs often remove BN layers because inaccurate BN statistics hinder exploitation, shallow ANNs can still train effectively without BN. For SNNs, however, removing BN can lead to notable performance drops because spiking neurons exhibit more sensitive activation and gradient dynamics [1]. Therefore, our goal is to design a **BN mechanism that enables effective RL training for SNNs**, rather than to propose a general ANN method.
>
> Although the proposed CaRe-BN can also implemented in ANNs, the motivation and improvements are all fundamentally SNN-specific. As discussed in Section 5.5:
> ```
> The results in Figure 6 (d) indicate that ANNs with CaRe-BN perform similarly to their baseline counterparts without CaRe-BN. This outcome is expected for the following reasons: (i) Shallow ANNs can already train stably and effectively without normalization, so adding CaRe-BN does not provide significant improvements. (ii) While CaRe-BN provides more precise estimates of BN statistics, this does not negatively impact the RL training process. These results further underscore that the improvements observed are not due to a stronger RL mechanism, but rather to the SNN-specific normalization strategies.
> ```
> We have updated both the Abstract and the Introduction to clearly state this scope and motivation (lines 019-020 and 080-083).
>
>
> >W3: Section 5.2 / Figure 3: The evidence in Figure 3 appears insufficient to conclude that the method yields more precise BN statistics, and contributes to better reward. We can only observe that CaRe-BN gets obviously higher expected rewards on three tasks out of five. It is unknown what is the root cause of such observation.
>
> Thank you for this question. In the revised version, we further measure the Wasserstein distance between the true activation distribution and the Gaussian moving distribution used inside BN layers, and we have added these results to Figure 3. As shown in the updated figure, CaRe-BN consistently maintains a distribution closer to the real activation distribution, enabling more accurate normalization and consequently more reliable policy outputs during exploration.
>
> We have added the more detailed discussions in Section 5.2.

---

> ### Author Response · Authors · 2025-11-19
> **Reply 2/3**
>
> >W4: How many time steps are used in the SNNs? How sensitive is performance to this hyperparameter, and what justifies the chosen value to be used in the baseline setting? Also, ablation study on the time step should be provided.
>
> Thank you for the question. All direct trained SNN models in our experiments use a fixed simulation time step of 5, following the standard setting adopted in prior SNN-based RL works [2–4].
>
>
> For the baseline setup, all hyperparameters related to the spiking actor network are kept identical to those in [2–4], while all RL-related hyperparameters match those used by their ANN counterparts. This ensures a fair and consistent comparison across all experiments.
>
> Regarding the ablation study on the SNN simulation time steps, Table R1 reports the performance across environments, as well as the average performance gain (APG) over the ANN baseline. We observe that SNNs generally benefit from larger simulation time steps, and CaRe-BN achieves even stronger results when using 8 SNN simulation steps (up to 6.32% improvement over ANNs). However, we report the main results using an SNN simulation time step of 5, following the standard configuration adopted in prior SNN-based RL studies [2–4].
>
> **Table R1.** Performance across environments and APG over ANN baselines for CaRe-BN under different SNN simulation time steps, where $\pm$ denotes one standard deviation.
> | SNN simulation time steps | IDP-v4   | Ant-v4   |HalfCheetah-v4|Hopper-v4|Walker2d-v4|APG|
> | -------- | :--------: | :--------: | :--------: | :--------: | :--------: | :----:|
> |$2$| $953 \pm 247$| $4924 \pm 171$|$7635 \pm 392$| $3588 \pm 10$| $3885 \pm 1365$| $-23.80$%|
> |$3$| $9285 \pm 100$| $5078 \pm 325$| $8190 \pm 567$| $3522 \pm 89$| $4391 \pm 282$|$2.03$%|
> |$5$| $9348 \pm 2$| $5373 \pm 159$| $9563 \pm 442$| $3586 \pm 49$| $4296 \pm 268$|$5.90$%|
> |$8$|$9354 \pm 1$|$5417 \pm 421$|$9989 \pm 278$|$3479 \pm 95$|$4311 \pm 348$|$6.32$%|
>
> We have added this table to Appendix D.5.6 in the revised version.
>
>
> >Q1: Figure 1: What does the “update step” represent—training parameter updates or an SNN inference time step?
>
> Thank you for pointing out this ambiguity. In Figure 1, the “update step” refers to training parameter update iterations, not SNN inference time steps. We have revised the figure caption and replaced “update step” with “update iteration” to avoid confusion in the updated version.
>
>
> >Q2: I could not find details of the network architectures used. Please describe the models (layers, widths, neuron types, readouts, etc.).
>
> We apologize for the confusion. We have now added the general framework of our model in Section 5.1, and the full network architectures and hyperparameters are provided in Appendix D.1 and Appendix D.3. Here is a summary of our continuous control experiments:
>
> 1. The spiking actor network has two layers with 256 spiking neurons per hidden layer.
> 2. CaRe-BN layers are inserted between every pair of adjacent layers in the spiking actor network.
> 3. We test multiple spiking neuron types, including LIF neurons, current-based LIF neurons (CLIF) [2], and dynamic neurons (DN) [3]. Detailed dynamics and parameter settings for each neuron model are shown in Appendix D.2 and Appendix D.3.
> 4. We use deterministic population encoding with Gaussian receptive fields and IF neurons to generate input spike trains (please refer to Appendix D.1.2 for details), following the prior works [2, 4].
> 5. The SNN output is produced by a single non-leaky, non-firing readout layer connected to the final SNN layer. We directly read the membrane potentials of these neurons as continuous outputs.
> 6. The spiking actor network is co-trained with a deep ANN critic network using STBP under the standard actor–critic framework in RL.

---

> ### Author Response · Authors · 2025-11-19
> **Reply 3/3**
>
> >Q3.1: How are SNN time steps integrated with the RL policy?
>
> Thank you for the question. The SNN simulation time steps are independent of the RL environment time step. For each RL environment step, the SNN agent performs one forward inference, which internally consists of 5 simulation time steps. After this inference, all internal neuron states (e.g., membrane potentials) are reset, meaning the SNN does not carry temporal dynamics across environment steps and does not accumulate information beyond the 5 step simulation window.
>
> We have now stated this explicitly in the Experiment Setup section (lines 340-342) in the revised version.
>
>
> >Q3.2: Do you aggregate/compress time steps to produce the actor/critic outputs?
>
> Thank you for this question. The detailed description is provided in the Appendix D.1.2, and we summarize it here for clarity.
>
> For the spiking actor network, we do not aggregate or compress spikes across time steps. Instead, after running the SNN for the fixed simulation window, we directly read out the membrane potential of a set of non-leaky, non-spiking decoder neurons attached to the final layer. This simple readout mechanism avoids additional floating point multiplications and preserves the energy efficiency of the SNN design.
>
> Besides, the critic network is a standard ANN, identical to the one used in the original RL algorithms.
>
>
>
> >Q3.3: Which components are SNNs and which components are not?
>
> Thank you for pointing this out. In our Actor–Critic framework, the actor network is an SNN, while the critic network remains an ANN. This hybrid design is the standard and most commonly used setting in SNN-based RL works [2–4]. The proposed CaRe-BN layers are inserted between every pair of adjacent layers in the spiking actor network.
>
> We have updated Section 5.1 to state this explicitly.
>
>
>
>
> **References**
>
> [1] Hanle Zheng, Yujie Wu, Lei Deng, Yifan Hu, and Guoqi Li. Going deeper with directly-trained larger spiking neural networks. In Proceedings of the AAAI conference on artificial intelligence, volume 35, pp. 11062–11070, 2021.
>
> [2]Guangzhi Tang, Neelesh Kumar, Raymond Yoo, and Konstantinos Michmizos. Deep reinforcement learning with population-coded spiking neural network for continuous control. In Conference on Robot Learning, pp. 2016–2029. PMLR, 2021.
>
> [3]Duzhen Zhang, Tielin Zhang, Shuncheng Jia, and Bo Xu. Multi-sacle dynamic coding improved spiking actor network for reinforcement learning. In Proceedings of the AAAI conference on artificial intelligence, volume 36, pp. 59–67, 2022.
>
> [4]Ding Chen, Peixi Peng, Tiejun Huang, and Yonghong Tian. Fully spiking actor network with intralayer connections for reinforcement learning. IEEE Transactions on Neural Networks and Learning Systems, 36(2):2881–2893, 2024a.

---

> ### Comment · Reviewer_Yg5y · 2025-11-21
>
> W1. Thank you for the revisions to the abstract and introduction, and for posting the updated content here. This addresses my concern.
>
> W2 .Thank you for the clarification; it makes the paper much clearer.
>
> W3. The additional quantitative results support the claim better.
>
> W4. The new results show that 𝑇=5 is a reasonable choice that balances reward and efficiency. The outcome is convincing, and I appreciate the quantitative evidence provided.
>
> Q1, Q2, Q3: Thank you for the further explanations and for revising the manuscript to better detail the model architecture. I have increased the soundness score as well as the overall score. Good rebuttal, and good luck with your paper!

---

> > ### Author Response · Authors · 2025-11-21
> > **Thank You**
> >
> > We sincerely thank you for your kind words and for raising the score. Please feel free to let us know if you have any further questions.

---

### Official Review · Reviewer_gu7R · 2025-11-01

**Soundness:** 2
**Presentation:** 3
**Contribution:** 2
**Rating:** 4
**Confidence:** 4

**Summary:**

The paper targets a practical instability in online RL with SNNs: BatchNorm (BN) statistics are hard to estimate under non-stationary data and replay, leading to train–inference mismatch and noisy updates. It proposes CaRe-BN, a two-part BN strategy: (i) Confidence-adaptive BN (Ca-BN), which linearly fuses the previous moving estimate and the current mini-batch estimate using variance-based gains (a Kalman-style, minimum-MSE update under stated assumptions); and (ii) Re-calibration (Re-BN), which periodically re-estimates inference statistics from larger replay samples to correct accumulated drift. CaRe-BN is drop-in and leaves the inference path unchanged. Integrated with TD3/DDPG and LIF/CLIF neurons, experiments on MuJoCo report consistent gains and, on average, SNN+CaRe-BN outperforming a vanilla ANN-TD3 baseline. The paper also argues that “more precise BN statistics” improve exploration quality.

**Strengths:**

1. Clear problem framing with a simple, principled fix. The paper isolates BN statistic mismatch as a key bottleneck in SNN-RL and proposes a plug-and-play estimator with an intuitive confidence-weighted fusion + periodic re-calibration backed by a tractable derivation.

2. Consistent empirical gains with informative ablations. Across several MuJoCo tasks and spiking neuron types, CaRe-BN improves stability/returns, and ablations suggest Ca-BN and Re-BN are complementary, supporting the design rationale.

3. Practical applicability and minimal engineering overhead. The method does not alter the inference graph, is easy to integrate into standard actor–critic pipelines, and is plausibly extensible beyond SNNs to other online/temporal settings where BN statistics drift.

**Weaknesses:**

1. Theory–data distribution mismatch in Theorem 1 (target inconsistency between moving BN and batch BN)
The theorem assumes that the “moving/prior” BN estimator and the “mini-batch/observation” BN estimator are unbiased estimates of the same target distribution with correctly specified variances. In online RL with replay, this assumption is typically violated: the moving BN statistics track the current inference distribution, while the batch BN statistics are computed from a replay mixture of (often older) policies, with temporal correlation and potentially heavy-tailed activations. Consequently, the two estimators do not share the same target, unbiasedness/i.i.d. fail, and the linear fusion with gain 𝐾 is not guaranteed to be valid or optimal (and may introduce bias).

2. Limited experimental scope beyond MuJoCo
Although the paper compares against common SNN normalization baselines (e.g., BNTT), the evaluation remains restricted to MuJoCo continuous control. To substantiate generality and the claimed benefits under stronger non-stationarity, the method should also be tested on exploration-hard and vision-based benchmarks (e.g., Atari).

3. Exploration claim lacks causal evidence and proper metrics
Section 5.2 infers “more precise BN statistics → better exploration” primarily from an “exploration return” curve without defining the metric, controlling confounders (action-noise schedule, value-estimation bias, policy smoothness), or reporting standard exploration measures (state-coverage/visitation, occupancy entropy, trajectory diversity, feature-space coverage). No experiments are provided on exploration-hard tasks or against classic intrinsic-motivation baselines (ICM, RND, NGU/Plan2Explore, count-based variants). The authors need to further analyze the reason why the proposed method can improve the efficiency of exploration.

4. Unestablished advantage over strong ANN-RL and unmeasured energy claims
The only ANN comparison uses a vanilla ANN-TD3 (in the appendix). Modern, stronger ANN-RL baselines (e.g., SAC; TD3/SAC with LN/GN/RMSNorm or Batch Renorm/AdaBN; distributional critics; entropy/parameter-noise regularization) are not evaluated, so superiority over ANN remains unproven. Moreover, the touted energy advantage of SNNs is theoretical and specific to neuromorphic hardware; no deployment-phase energy/latency measurements (or credible proxies) are reported, nor is training vs. inference cost disentangled. Open discussion: beyond theoretical energy efficiency on neuromorphic hardware, what concrete advantages does SNN + CaRe-BN have over strong ANN-RL methods (e.g., sample efficiency under non-stationarity, stability under distribution shifts, low-latency control, small-batch robustness), and can the authors provide controlled evidence for them?

**Questions:**

1. Theory & Theorem 1 assumptions (target inconsistency and variance modeling).
Could you formally define the target distribution for BN statistics in online RL and quantify the mismatch between moving BN (inference) and batch BN (replay) (e.g., TV/Wasserstein distance over activations)? Please provide bias/variance bounds for your fused estimator under distribution shift, report effective sample size N_"eff" with autocorrelation estimates for replayed batches, and clarify how you separate process noise vs. observation noise when computing the confidence gains (consider an adaptive Kalman/EM treatment instead of a single residual EMA).
2.“Better exploration” claim: metrics, confounders, and causal evidence.
What is the exact definition of exploration return? To establish causality, could you (i) align random seeds and action-noise schedules, (ii) report standard exploration metrics—state coverage/visitation counts (or pseudo-counts), occupancy entropy, trajectory diversity, feature-space coverage, and (iii) add Actor-only vs. Critic-only CaRe-BN ablations and noise-aligned runs? Please also include tests on exploration-hard benchmarks (e.g., Atari) and comparisons to intrinsic-motivation baselines (ICM, RND).
3.External validity beyond MuJoCo and advantage over strong ANN-RL.
Beyond MuJoCo, can you evaluate on vision-based and non-stationary settings (context/reward switches, parameter drift) and compare against stronger ANN baselines (e.g., SAC; TD3/SAC with LN/GN/RMSNorm or Batch Renorm/AdaBN; distributional critics; entropy/parameter-noise regularization)? If the paper claims energy advantages, please report deployment-phase energy/latency (or credible proxies/ simulation power consumption estimation) and disentangle training vs. inference costs; otherwise, consider narrowing the energy-efficiency claims.

---

> ### Author Response · Authors · 2025-11-19
> **Reply 1/4**
>
> Thank you for the detailed reviewing and valuable suggestions. We would like to address your concerns as follows.
>
> > W1: Theory–data distribution mismatch in Theorem 1 (target inconsistency between moving BN and batch BN) The theorem assumes that the “moving/prior” BN estimator and the “mini-batch/observation” BN estimator are unbiased estimates of the same target distribution with correctly specified variances. In online RL with replay, this assumption is typically violated: the moving BN statistics track the current inference distribution, while the batch BN statistics are computed from a replay mixture of (often older) policies, with temporal correlation and potentially heavy-tailed activations. Consequently, the two estimators do not share the same target, unbiasedness/i.i.d. fail, and the linear fusion with gain 𝐾 is not guaranteed to be valid or optimal (and may introduce bias).
>
> Thank you for your thoughtful question. We agree that the state distribution in the replay buffer (used during training) differs from the state distribution under the current policy (during inference). However, this mismatch does **not** affect the validity of CaRe-BN, as CaRe-BN’s BN fusion operates solely on the replay buffer distribution.
>
> During training, the CaRe-BN layers in the spiking actor are optimized only to maximize the critic’s evaluation over samples drawn from the replay buffer. This optimization objective
> $J(\phi)=\mathbb{E} _ {s\in \text{replay}}Q _ \theta^{\pi}(s,\pi _ \phi(s))$ depends exclusively on the replay distribution (where $\pi _ \phi$ is the actor network paramized by $\phi$ and $Q _ \theta^{\pi}$ is the critic network paramized by $\theta$ ). As shown in Algorithm 1, the moving BN statistics are updated only during training, using mini-batches sampled from the same replay buffer. Therefore, the two estimators share the same target distribution, and the linear fusion in CaRe-BN is well-defined in this setting.
>
>
> > W2: Limited experimental scope beyond MuJoCo Although the paper compares against common SNN normalization baselines (e.g., BNTT), the evaluation remains restricted to MuJoCo continuous control. To substantiate generality and the claimed benefits under stronger non-stationarity, the method should also be tested on exploration-hard and vision-based benchmarks (e.g., Atari).
>
> Thank you for this valuable suggestion. We have added a new set of experiments on Atari 2600 games, including Pong-v5, Breakout-v5, Freeway-v5, Seaquest-v5, and SpaceInvaders-v5. Due to the limited time and computational resources during the rebuttal period, these experiments were conducted using RAM-based input and evaluated over 3 random seeds. The experimental settings are described in Section 5.1 (lines 323-327), and the detailed hyperparameters are provided in Appendix D.3.
>
> We include the learning curves for SNNs with and without CaRe-BN in Figure 7. SNN agents equipped with CaRe-BN achieve substantially better performance. These results demonstrate that CaRe-BN not only improves SNN-RL in continuous control scenarios but also generalizes effectively to exploration-hare and discrete-action environments.
>
>
> > W3.1: Exploration claim lacks causal evidence and proper metrics Section 5.2 infers “more precise BN statistics → better exploration” primarily from an “exploration return” curve without defining the metric, controlling confounders (action-noise schedule, value-estimation bias, policy smoothness), or reporting standard exploration measures (state-coverage/visitation, occupancy entropy, trajectory diversity, feature-space coverage).
>
> In Section 5.2, the term “exploration return” refers to the average episodic return obtained during training, i.e., the sum of rewards collected in a single rollout while the agent interacts with the environment. During these interactions, the agent selects actions using its current policy plus the exploration noise defined by the underlying RL algorithm. For TD3 (used in Section 5.2), this corresponds to a Gaussian noise $\mathcal{N}(0,0.1)$, as reported in Appendix D.3. All exploration-related components and hyperparameters are kept identical across methods to ensure a fair comparison.
>
> We have revised Section 5.2 to avoid any confusion. We also further address this point in our responses to the next question.

---

> ### Author Response · Authors · 2025-11-19
> **Reply 2/4**
>
> > W3.2: No experiments are provided on exploration-hard tasks or against classic intrinsic-motivation baselines (ICM, RND, NGU/Plan2Explore, count-based variants). The authors need to further analyze the reason why the proposed method can improve the efficiency of exploration.
>
> **Exploration-hard tasks:** We have added experiments on Atari 2600 games, as discussed in our response to Weakness 2. These results further demonstrate that CaRe-BN maintains stable performance even in harder exploration settings.
>
> **Intrinsic-motivation baselines:** We would like to clarify that the goal of CaRe-BN is not to introduce an intrinsic motivation mechanism or to boost exploration in the way that ICM, RND, NGU/Plan2Explore, and count-based methods do. Instead, CaRe-BN aims to provide accurate and stable normalization statistics that do not harm exploration, addressing the issues caused by traditional BN in RL. Therefore, we respectfully disagree that we should compare the CaRe-BN with these approches, as we do not claim to outperform such exploration-enhancing mechanisms.
>
>
> **Further analyze the reason:** We measure the Wasserstein distance between the true activation distribution and the Gaussian moving distribution used inside BN layers and added the results to Figure 3 in the revised version. As shown in Figure 3, CaRe-BN maintains a distribution closer to the real activation distribution, resulting in more accurate normalization and consequently more reliable policy outputs during exploration.
>
> Thank you for the helpful suggestions. We have updated the revision accordingly.
>
> > W4.1: Unestablished advantage over strong ANN-RL and unmeasured energy claims The only ANN comparison uses a vanilla ANN-TD3 (in the appendix). Modern, stronger ANN-RL baselines (e.g., SAC; TD3/SAC with LN/GN/RMSNorm or Batch Renorm/AdaBN; distributional critics; entropy/parameter-noise regularization) are not evaluated, so superiority over ANN remains unproven.
>
> As for stronger ANN-RL baselines, we added comparisons using SAC, a stronger modern off-policy RL algorithm. We report both the ANN-SAC and SNN-SAC (with and without CaRe-BN) results. As shown in Figure 9 in Appendix D.4.1, SNNs equipped with CaRe-BN also surpass their ANN counterparts under SAC, further supporting the claim that CaRe-BN benefits SNN-RL beyond the TD3 setting.
>
> As for Rl algorithm with LN/GN/RMSNorm/Batch Renorm/AdaBN, we would like to clarify that these normalization techniques are typically not used in standard continuous-control RL pipelines for ANNs. Existing open source and widely adopted implementations of TD3/SAC do not include LN, GN, RMSNorm, Batch Renorm, or AdaBN, as shallow ANNs generally train reliably without such normalization layers. Consequently, there are no established RL baselines built upon these normalization methods in the ANN-RL literature.
>
> Thank you for this question. We have added the new SAC comparisons in the revised version to further demonstrate the potential advantages of CaRe-BN over ANN baselines.
>
> > W4.2: Moreover, the touted energy advantage of SNNs is theoretical and specific to neuromorphic hardware; no deployment-phase energy/latency measurements (or credible proxies) are reported, nor is training vs. inference cost disentangled.
>
> We have added experiments measuring both training time costs and inference energy costs in Appendix D.5. In summary:
> * CaRe-BN introduces **no** noticeable overhead in training time or memory compared to other BN variants. Its training speed and memory usage remain comparable to standard BN, ensuring the method does not incur additional training-phase cost.
> * SNNs equipped with CaRe-BN consume 16.48 nJ per inference on average, while ANN agents consume 1775.36 nJ on average, demonstrating the substantial inference energy advantage of SNNs.

---

> ### Author Response · Authors · 2025-11-19
> **Reply 3/4**
>
> > W4.3: Open discussion: beyond theoretical energy efficiency on neuromorphic hardware, what concrete advantages does SNN + CaRe-BN have over strong ANN-RL methods (e.g., sample efficiency under non-stationarity, stability under distribution shifts, low-latency control, small-batch robustness), and can the authors provide controlled evidence for them?
>
> Thank you for this insightful question. We agree that this is an important open problem for the broader SNN community: **beyond energy efficiency and biological plausibility, what advantages can SNNs offer over ANNs?**
>
> In the field of reinforcement learning, this work provides several potential advantages of SNNs equipped with CaRe-BN:
> 1. Improved sample efficiency and higher final performance (in lines 482-503).
> 2. Lower performance variance and better training stability (in lines 423-426).
>
> While these observations indicate potential advantages beyond energy efficiency, we believe that the full scope of SNN benefits (e.g., robustness to distribution shifts, low-latency control, and small-batch stability) remains a rich area for future exploration. We sincerely welcome further insights from you and other reviewers on this important open question.
>
> >Q1.1: Theory & Theorem 1 assumptions (target inconsistency and variance modeling). Could you formally define the target distribution for BN statistics in online RL and quantify the mismatch between moving BN (inference) and batch BN (replay) (e.g., TV/Wasserstein distance over activations)?
>
> Thank you for this question. The target distribution for BN statistics in online RL corresponds to the activation distribution induced by the states distributed in **replay buffer**, which is discussed in our response to Weakness 1.
>
> To quantify the mismatch between approximated moving BN statistics and real BN statistics from replay samples, we have added a new experiment in Section 5.2. Specifically, we compute the Wasserstein distance between the true activation distribution and the estimated distribution. The results in Figure 3 show that CaRe-BN consistently produces statistics that are closer to the real distribution, which is also discussed in our response to Weakness 3.2.
>
> >Q1.2: Please provide bias/variance bounds for your fused estimator under distribution shift, report effective sample size N_"eff" with autocorrelation estimates for replayed batches, and clarify how you separate process noise vs. observation noise when computing the confidence gains (consider an adaptive Kalman/EM treatment instead of a single residual EMA).
>
> We guess there may be a misunderstanding. CaRe-BN always estimates statistics using batches sampled directly from the replay buffer, meaning that the data distribution and target distribution are identical. As discussed in our response to Weakness 1, CaRe-BN does not fuse statistics coming from two different distributions.
>
> **As for the estimation bias/variance:** Figure 3 reports the Wasserstein distance between the true activation distribution and the estimated BN distribution, with shaded regions indicating standard deviations across runs. This directly reflects the estimator’s error behavior.
>
> **As for effective sample size:** Since samples are drawn i.i.d. from the replay buffer (the target distribution), there is no autocorrelation, and the effective sample size equals the batch size $N_{eff}=N$.
>
> **As for separating process noise and observation noise:** We approximate the process noise using the squared residual error between the current observation and the previous estimate. Because this contains both process and observation noise, we apply an EMA smoother to partially isolate observation noise form it and stabilize the estimation.
>
> **As for the reason why not using adaptive Kalman/EM:** We agree that a full adaptive Kalman with EM could yield more precise estimates. However, such methods require recurrent updates over multiple historical iterations, significantly increasing memory and compute cost, which is almost impracticable in online RL settings where BN statistics is updated millions of times. Using the lightweight CaRe-BN is a practical balance, avoiding heavy computation while reducing BN estimation error, as demonstrated in our experiments.

---

> ### Author Response · Authors · 2025-11-19
> **Reply 4/4**
>
> >Q2.1: “Better exploration” claim: metrics, confounders, and causal evidence. What is the exact definition of exploration return?
>
> The term “exploration return” refers to the average episodic return obtained during training, i.e., the sum of rewards collected in a single rollout while the agent interacts with the environment.
>
> >Q2.2: To establish causality, could you (i) align random seeds and action-noise schedules, (ii) report standard exploration metrics—state coverage/visitation counts (or pseudo-counts), occupancy entropy, trajectory diversity, feature-space coverage, and (iii) add Actor-only vs. Critic-only CaRe-BN ablations and noise-aligned runs?
>
> (i) All experiments are conducted using the same set of 5 random seeds across all methods for a fair comparison. The exploration noise follows the standard TD3 setting, a Gaussian noise $\mathcal{N}(0,0.1)$, which is identically applied across all runs.
>
> (ii) We have added additional analysis and clarifications in Section 5.2, including a more detailed discussion of how CaRe-BN affects the distributional accuracy of activation statistics, which leads to better exploration than other BN variants. As clarified in our response to Weakness 3.2, this parts mainly focus on the metric of distributional alignment, rather than the exploration metrics.
>
> (iii) We would like to clarify that CaRe-BN is applied only to the spiking actor network. The critic remains an ANN and is never used during inference, so no normalization modification is applied there. We have explicitly clarified this in the revised manuscript.
>
> >Q2.3: Please also include tests on exploration-hard benchmarks (e.g., Atari) and comparisons to intrinsic-motivation baselines (ICM, RND).
>
> Thank you for the suggestion. We have added experiments on Atari environments in the revised manuscript, as discussed in our response to Weakness 2. Additionally, we have discussed intrinsic-motivation baselines in the response to Weakness 3.2, clarifying that CaRe-BN is designed to preserve stable exploration rather than to actively boost it.
>
>
> >Q3.1: External validity beyond MuJoCo and advantage over strong ANN-RL. Beyond MuJoCo, can you evaluate on vision-based and non-stationary settings (context/reward switches, parameter drift) and compare against stronger ANN baselines (e.g., SAC; TD3/SAC with LN/GN/RMSNorm or Batch Renorm/AdaBN; distributional critics; entropy/parameter-noise regularization)?
>
> Thank you for the suggestion. We have added experiments on Atari environments to further demonstrate the generalizability of CaRe-BN beyond MuJoCo, as detailed in our response to Weakness 2.
>
> As for stronger ANN baselines, we also evaluated our method under the SAC algorithm, which further supports the potential advantages of CaRe-BN over ANN baselines.
>
> As for the TD3/SAC variants with LN/GN/RMSNorm or Batch Renorm/AdaBN, we clarify (as discussed in our response to Weakness 4.1) that such baselines do not exist in the RL literature, as standard actor–critic networks are shallow enough to train reliably without these normalization layers.
>
> >Q3.2: If the paper claims energy advantages, please report deployment-phase energy/latency (or credible proxies/ simulation power consumption estimation) and disentangle training vs. inference costs; otherwise, consider narrowing the energy-efficiency claims.
>
> Thank you for this suggestion. We have added experiments measuring both training time costs and inference energy consumption in Appendix D.5. As discussed in our response to Weakness 4.2, SNNs equipped with CaRe-BN introduce no noticeable training overhead, while providing a substantial inference energy advantage over ANN counterparts.

---

> > ### Comment · Reviewer_gu7R · 2025-11-26
> >
> > The author provided further theoretical explanations, stronger experimental comparison results with ANN baselines, and also demonstrated the energy efficiency advantages of the proposed method. This greatly resolved my confusion, and therefore, I will increase my score.

---

> > > ### Author Response · Authors · 2025-11-26
> > > **Thank You**
> > >
> > > We sincerely appreciate your thoughtful feedback and the updated score. Please feel free to let us know if you have any further questions.

---

### Meta-Review · Area_Chair_DerV · 2026-01-06

**Summary:**

This paper addresses a well-known but under-explored challenge in spiking neural network reinforcement learning (SNN-RL): the instability and mismatch of Batch Normalization (BN) statistics under online, non-stationary training with replay buffers. The authors propose CaRe-BN, a principled and lightweight solution combining confidence-adaptive BN fusion (Ca-BN) and periodic recalibration (Re-BN).
Across the review process, reviewers initially raised concerns regarding (i) theoretical assumptions behind the BN fusion, (ii) limited experimental scope beyond MuJoCo, (iii) insufficient causal analysis of the exploration claim, (iv) unclear positioning with respect to ANN-RL baselines, and (v) missing efficiency and ablation studies.
During the rebuttal and revision period, the authors provided extensive clarifications, added new theoretical explanations, substantially expanded experiments (including Atari 2600, SAC baselines, energy measurements, neuron-model generalization, and SNN timestep ablations), and revised the manuscript to better clarify scope and claims. Importantly, all reviewers explicitly acknowledged that their major concerns were addressed and increased their scores, with one initially critical reviewer (gu7R) stating that the rebuttal “greatly resolved my confusion.”
Overall, the discussion converged to a consensus that the paper presents a solid, well-motivated contribution to SNN-RL, with convincing empirical evidence and clear practical relevance.

**Reviewer Concerns:**

Concerns addressed by the rebuttal:
- Theoretical validity of BN fusion (gu7R, EiPv):
The authors clarified that CaRe-BN operates entirely on the replay-buffer distribution during training, resolving the concern that the fused estimators target different distributions. Additional derivations and explanations of variance modeling and confidence estimation were added, and reviewers acknowledged the clarification.
- Limited evaluation scope (gu7R, eBqg):
New experiments on Atari 2600 games demonstrated effectiveness beyond MuJoCo, including discrete-action and more exploration-challenging settings.
- Exploration claim and causal analysis (gu7R, Yg5y):
The authors clarified the definition of “exploration return,” aligned experimental conditions, and added Wasserstein-distance analysis showing improved alignment between estimated and true activation distributions, providing a plausible mechanistic explanation.
- ANN baseline comparisons (gu7R):
Stronger ANN baselines (SAC) were added, showing that SNNs with CaRe-BN remain competitive or superior, while also clearly narrowing the claim that the main contribution is SNN-specific.
- Efficiency, overhead, and neuron generality (eBqg, EiPv):
Training time, memory usage, inference energy, and experiments with advanced spiking neuron models were added, showing minimal overhead and consistent gains.
- Clarity and presentation issues (Yg5y):
Claims about BN and SNNs were revised, architecture details and timestep ablations were added, and ambiguities in figures and experimental setup were resolved.

Remaining concerns:
Some broader questions (e.g., formal bias/variance bounds under arbitrary distribution shift, or comparisons to intrinsic-motivation exploration methods) remain open, but these are acknowledged as future work and no longer viewed as blockers for acceptance.

**Reviewer Scores:**

Reviewer Scores
Based on rebuttal responses and explicit reviewer comments:

Reviewer gu7R: 4  (explicitly stated confusion resolved)

Reviewer Yg5y: 2  (major concerns addressed, soundness and overall score increased)

Reviewer EiPv: 6  (clarifications and revisions led to score increase)

Reviewer eBqg: 6 (new experiments fully addressed weaknesses)

The post-discussion consensus score (attitude of the four reviewers) is clearly above the acceptance threshold.

---

### Decision · Program_Chairs · 2026-01-26

Accept (Poster)